# Reticulate allopolyploidy and subsequent dysploidy drive evolution and diversification in the cotton family

Ren-Gang Zhang [1,2,12], Hang Zhao[3,4,12], Justin L. Conover [5,6,12], Hong-Yun Shang [1,2,12], De-Tuan Liu[1], Min-Jie Zhou[1,2], Xiong-Fang Liu [1], Kai-Hua Jia [7], Shi-Cheng Shao[8], Meng-Meng Li[3], Chong-Yang Jin[3], Yi-Hui Liu[4], Xiao-Yi Shen[3], Da-Wei Li[9] ✉, Martin A. Lysak [10] ✉, Jonathan F. Wendel [11] ✉, Xiao-Yang Ge [3] ✉ & Yong-Peng Ma [1] ✉

Polyploidy and subsequent post-polyploid diploidization (PPD) are key drivers of plant genome evolution, yet their contributions to evolutionary success remain debated. Here, we analyze the Malvaceae family as an exemplary system for elucidating the evolutionary role of polyploidy and PPD in angiosperms, leveraging 11 high-quality chromosome-scale genomes from all nine subfamilies, including newly sequenced, near telomere-to-telomere assemblies from four of these subfamilies. Our findings reveal a complex reticulate paleoallopolyploidy history early in the diversification of the Malvadendrina clade, characterized by multiple rounds of species radiation punctuated by ancient allotetraploidization (Mal-β) and allodecaploidization (Mal-α) events around the Cretaceous–Paleogene (K–Pg) boundary. We further reconstruct the evolutionary dynamics of PPD and find a strong correlation between dysploidy rate and taxonomic richness of the paleopolyploid subfamilies ($R^2 \geq 0.90$, $P < 1e\text{-}4$), supporting the "polyploidy for survival and PPD for success" hypothesis. Overall, our study provides a comprehensive reconstruction of the evolutionary history of the Malvaceae and underscores the crucial role of polyploidy–dysploidy waves in shaping plant biodiversity.

Polyploidy, or whole-genome duplication (WGD), is a pivotal phenomenon in eukaryotic evolution, particularly in flowering plants[1]. It is widely accepted that all angiosperms have undergone two common ancient polyploidy events (i.e., paleopolyploidy) in their evolutionary past[2], with hundreds more occurring across the major clades of angiosperms[3]. In addition, reticulate allopolyploidization, involving recurrent hybridization with diploid or polyploid genomes, produces coexisting genetically related allopolyploids, which is likely a key process in the origin and evolution of plant lineages[4–6]. The prevalence of such reticulations challenges the adequacy of bifurcating tree models to represent the diversification and evolutionary history of angiosperms, and suggests that the use of network-based approaches

can be more appropriate[6,7]. Studies using whole-genome data have explored reticulate allopolyploidization at the genus (e.g., poppies[4]) and subfamily (bamboos[8]) levels. However, the lack of high-quality genome assemblies and limited methods for tracing the complex evolution of ancient polyploids hinder our understanding of how reticulate allopolyploidizations drive plant diversity at higher taxonomic levels (e.g., families).

Polyploidy has been linked to survival[9], adaptation[10,11], speciation, diversification and radiation[12–15] as well as evolutionary novelty and key innovations[16], positioning it as a potential driver of evolutionary success in angiosperms[16]. However, evidence suggests that newly formed polyploids (i.e., neopolyploids) face higher extinction rates and lower

A full list of affiliations appears at the end of the paper. ✉e-mail: lidawei@wbgcas.cn; martin.lysak@ceitec.muni.cz; jfw@iastate.edu; gexiaoyang@caas.cn; mayongpeng@mail.kib.ac.cn

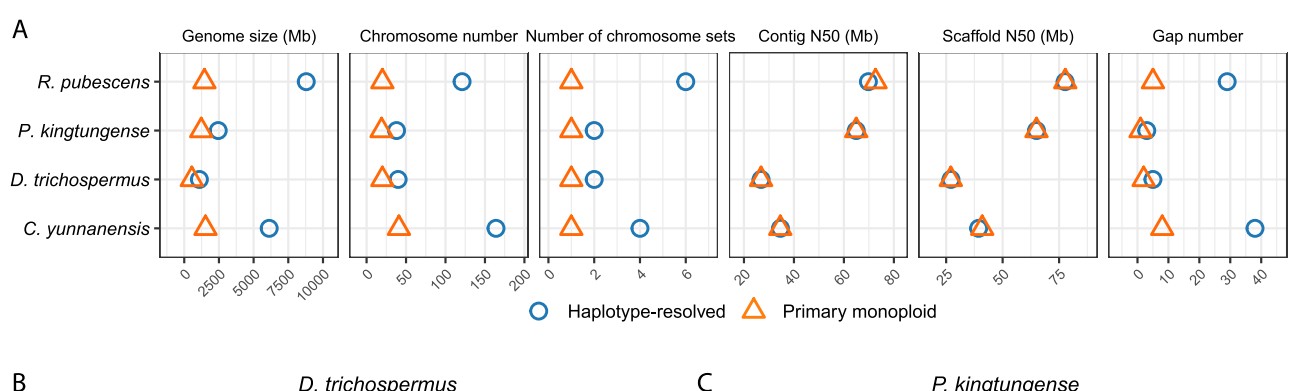

**Fig. 1 | Genome features of four species in Malvaceae. A** Summary of genome assemblies in this study (see Supplementary Table 2 for more details). **B**–**E** Genomic characteristics visualized via circos plots for *D. trichospermus* (**B**), *P. kingtungense* (**C**), *C. yunnanensis* (**D**), and *R. pubescens* (**E**). Circles denote **a** karyotype, colored by subgenome assignments, **b** Class I TE density, **c** Class II TE density, **d** PCG density, **e** tandem repeat content, **f** GC content and **g** inparalogous syntenic blocks, colored by alignments to proto-chromosomes of tPMK.

speciation rates than their diploid relatives[17,18], perhaps due to evolutionary disadvantages such as reduced fertility and fitness, loss of self-incompatibility and the initial demographic hurdle[19,20]. This has led some to argue that polyploids are rarely successful[19], creating a paradox between the evolutionary significance of polyploidy and the notion of polyploids as evolutionary dead-ends.

Observations of significant 'lag-times' between paleopolyploidy events and subsequent species radiations in major angiosperm clades[13] have prompted hypotheses to reconcile this paradox. Dodsworth and colleagues proposed that post-polyploidization diploidization (PPD)—an often-overlooked process—may be crucial to the evolutionary success of angiosperm lineages, rather than polyploidy per se[21]. Specifically, Mandáková & Lysak argued that PPD could drive speciation and cladogenesis through dysploidy changes (i.e., fixed changes of chromosome number), leading to intraspecific genetic diversity, reproductive isolation, speciation and ultimately the diversification of large angiosperm groups[22]. However, these hypotheses remain largely untested.

The cotton family (Malvaceae s.l.) holds significant ecological and economic importance, with many members integral to tropical and subtropical ecosystems and some others domesticated as crops, such as cotton (*Gossypium* spp.), cacao (*Theobroma cacao*) and durian (*Durio zibethinus*). The family comprises nine subfamilies: Bombacoideae, Brownlowioideae, Byttnerioideae, Dombeyoideae, Grewioideae, Helicteroideae, Malvoideae, Sterculioideae, and Tilioideae[23,24]. Phylogenetically, Byttnerioideae and Grewioideae form the Byttneriina clade, while the remaining seven subfamilies constitute a larger Malvadendrina clade[23–26]. Within Malvadendrina, Bombacoideae and Malvoideae unite as the Malvatheca clade. However, the relationships between Malvatheca and the other five subfamilies remain unresolved due to substantial phylogenetic discordance[27–33]. Beyond the phylogenetic conflicts, the paleopolyploid history within the Malvaceae is complex and remains largely uncertain[33–38] (see detailed reviews in Supplementary Note 1), potentially fueling the observed phylogenetic discordance. Given the diverse lineages in Malvaceae, this family may also provide an ideal model for testing the hypotheses surrounding the evolutionary paradox of polyploidy.

Here, to unveil the complex evolutionary history of the family, we compile high-quality genome assemblies of representative species from all nine subfamilies, including four newly sequenced genomes from Brownlowioideae, Dombeyoideae, Helicteroideae and Tilioideae. Leveraging updated genomic data and advanced subgenome-aware analyses, we resolve the 'Malvaceae mystery'[33]—a puzzle shaped by complex paleoallopolyploidizations (allotetraploidy, allohexaploidy and allodecaploidy) and multiple radiations with incomplete lineage sorting (ILS) and introgression/hybridization within the Malvadendrina clade. Moreover, we find evidence supporting the positive correlation between the extent of post-polyploid dysploidy and the evolutionary success of paleopolyploids. Taken together, reticulate allopolyploidizations and subsequent genome diploidizations likely trigger multiple rounds of radiation and facilitate the survival and evolutionary success of the Malvadendrina paleopolyploids.

## Results and discussion
### Chromosome-scale assemblies and annotations of Malvaceae genomes
To encompass all nine subfamilies of the Malvaceae, we sequenced and assembled chromosome-scale genomes for four species: *Diplodiscus trichospermus* ($2n = 2x = 40$, Brownlowioideae), *Pterospermum kingtungense* ($2n = 2x = 38$, Dombeyoideae), *Craigia yunnanensis* ($2n = 4x = 164$, neoautotetraploid, Tilioideae) and *Reevesia pubescens* ($2n = 6x = 120 + 1$, neoautohexaploid with aneuploidy, Helicteroideae) (Fig. 1). We generated an extensive dataset comprising 419 Gb PacBio long HiFi reads (average depth 85×), 415 Gb short reads (104×), 1,479 Gb Hi-C reads (280×), and 75 Gb Nanopore Iso-seq reads, for

genome assembly and annotation (Supplementary Table 1). Due to high heterozygosity and/or autoneopolyploidy (Supplementary Figs. 1 and 2), we assembled the complete chromosome sets (haplotype-resolved) of the genomes and selected the primary (monoploid) set for further analyses (Fig. 1, Supplementary Figs. 3–8, Supplementary Table 2). The sizes of the complete genomes ranged from 1.1 (*D. trichospermus*) to 8.8 Gb (*R. pubescens*), whereas the monoploid genomes ranged from 0.5 (*D. trichospermus*) to 1.5 Gb (*C. yunnanensis*) (Fig. 1A, Supplementary Table 2), aligning with the estimates based on *k*-mer profiles (Supplementary Fig. 1). The monoploid assemblies comprised 19–41 pseudochromosomes, with one to eight gaps (Fig. 1, Supplementary Fig. 9, Supplementary Table 2). The contig N50 values ranged from 26.8 to 72.6 Mb, and the scaffold N50 values from 27.1 to 77.7 Mb (Fig. 1A, Supplementary Table 2), demonstrating high continuity. Comparison of *k*-mer spectra[39] against HiFi reads confirmed the assembly consistency (Supplementary Fig. 8). BUSCO[40] assessments based on the embryophyta_odb10 database indicated 98.9% to 99.4% completeness (Supplementary Table 2). Telomeric $(TTTAGGG)_n$ repeats were detected in most (92.5–97.4%) of the chromosomal ends (Supplementary Fig. 9), indicating the assemblies being nearly telomere-to-telomere. We also identified 33,126 to 57,219 protein-coding genes (PCGs) in the four genomes, with BUSCO completeness of 97.2 to 98.1% indicating robust annotation quality (Supplementary Table 3). Transposable elements (TEs) accounted for 55–70% of the genomes, with long terminal repeat retrotransposons (LTR-RTs) being the most prevalent, constituting 26–55% of the genomes (Supplementary Table 4). Plastid (159–163 kb, 78–79 PCGs) and mitochondrial (533–826 kb, 36–37 PCGs) genomes were also assembled for each species (Supplementary Table 5).

Complementing these newly sequenced genomes, we incorporated seven existing chromosome-scale assemblies: *Microcos paniculata*[41] (Grewioideae), *Theobroma cacao*[42] (Byttnerioideae), *Firmiana major*[43] (Sterculioideae), *Durio zibethinus*[36] (Helicteroideae), *Ochroma pyramidale*[44] (Bombacoideae), *Bombax ceiba*[45] (Bombacoideae) and *Gossypium raimondii*[34] (Malvoideae) (Supplementary Tables 6 and 7). Collectively, these 11 genomes represent all nine subfamilies of Malvaceae, with dual representatives from Helicteroideae and Bombacoideae, to address previously reported intra-subfamilial phylogenetic inconsistencies[24,27,29]. Our phylogenetic analyses show substantial nucleocytoplasmic conflicts and gene tree discordance within the Malvaceae, with the exceptions of the crown groups of Malvaceae, Malvadendrina and Malvatheca (Fig. 2A, Supplementary Fig. 10; Supplementary Note 2), underscoring the need to first dissect the complex paleopolyploid history for the Malvaceae to resolve these discrepancies.

### Complex reticulated paleoallopolyploidy within Malvadendrina
To resolve the paleopolyploid history, we first determined the relative ploidy (i.e, orthologous synteny depth to a reference genome; denoted as *p*)[46] for the 11 Malvaceae genomes through comparisons to the grape and cacao genomes, which have not undergone lineage-specific polyploidy since the paleohexaploid γ event shared by core eudicots[47,48]. In comparison to grape and cacao, all Malvadendrina genomes display orthologous synteny depth ranging from twofold to fivefold (i.e, $p = 2$ to 5; Fig. 2B, Supplementary Fig. 11), indicating that each species in the seven Malvadendrina subfamilies has experienced at least one polyploidization event (Fig. 3), as also evidenced by the synonymous substitution rate (*K*s) peaks (0.2–0.5) in paralogous syntenic genes (Fig. 2C). Specifically, *F. major*, *D. trichospermus*, *P. kingtungense* and *R. pubescens* showed a 2:1 orthologous synteny ratio relative to the cacao genome ($p = 2$; Fig. 2C, Supplementary Fig. 11A–D), signaling paleotetraploidy event(s) (Fig. 3). *Du. zibethinus* showed a 3:1 ratio ($p = 3$; Fig. 2C, Supplementary Fig. 11E), suggesting a paleohexaploidy event (Fig. 3) and confirming the previous conclusion[37]. *C. yunnanensis* showed a 4:1 ratio with a single in-

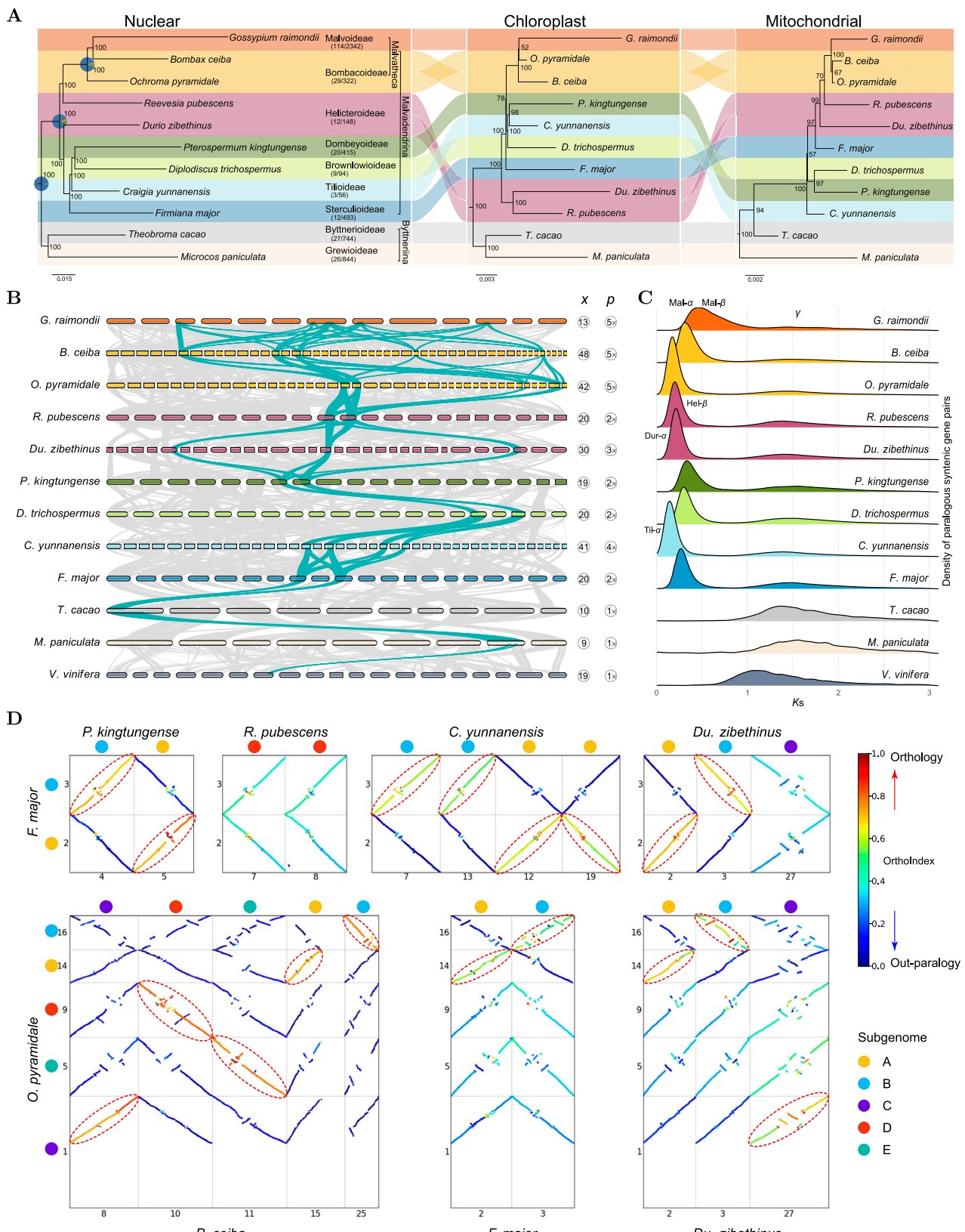

paralogue $K$s ($K$s = -0.2) peak ($p$ = 4; Fig. 2C, Supplementary Fig. 11F), suggesting either two sequential paleotetraploidy events or one paleooctoploidization event (Fig. 3). The genomes of *O. pyramidale*, *B. ceiba* and *G. raimondii* exhibited a 5:1 ratio ($p$ = 5; Fig. 2C, Supplementary Fig. 11G–I), indicative of paleodecaploidy event(s) (Fig. 3), aligning with previous studies[35,37,38,49].

We then investigated whether these polyploidy events were shared among Malvadendrina subfamilies, using orthologous synteny patterns[46] (Supplementary Figs. 12–14) and chromosome- to subgenome-scale phylogenies[5] (Supplementary Figs. 15–19). Pairwise comparisons showed consistent 1:1 orthologous synteny patterns between the three paleotetraploid genomes of *F. major*, *D.*

**Fig. 2 | Phylogenetic relationships and paleopolyploidy events in the Malvaceae. A** Comparative phylogenetic trees of the Malvaceae based on proteincoding genes from the nuclear genome (16,370 multi-copy by ASTRAL and 1904 single-copy genes by IQ-TREE yielding the same topology and all 100% support values), the chloroplast genome (79 genes) and the mitochondrial genome (37 genes). The trees were rooted by two outgroups (*Vitis vinifera* and *Carica papaya*). The branch lengths represent the nucleotide substitutions per site, and the values in the nodes represent the bootstrap percentages from IQ-TREE. The confidently well-resolved nodes are indicated by pie charts with the frequencies of three gene tree topologies (q1, q2 and q3; q1 » q2/q3) calculated in ASTRAL (see Supplementary Fig. 10 for more details). The numbers in brackets indicate the number of genera and species in the given subfamily. **B** Orthologous synteny between the Malvaceae genomes and the grapevine genome (*V. vinifera*). Monoploid chromosome number (*x*) and relative ploidy level (*p*) are given for each species. *p* > 1 indicates polyploidization events (e.g., 2× indicating paleotetraploidy) since the divergence of the Malvaceae genomes from the Byttneriina genomes (*M. paniculata* and *T. cacao*; *p* = 1). Syntenic blocks exemplifying the relative ploidy patterns are highlighted in green. **C** *K*s distribution profiles for paralogous syntenic gene pairs within each species. All genomes share a peak corresponding to the γ paleohexaploidy event. **D** Illustrative dot plots colored by Orthology Index to show the patterns of orthologous synteny between representative chromosomes in Malvadendrina paleopolyploids. Both the *x*- and *y*-axes in each subplot represent the representative chromosomes from two representative species. The intergenomic syntenic blocks with a relatively high Orthology Index indicate the orthologous relationships, as highlighted by dashed ellipses. These dot plots show orthologous synteny ratios of 1:1 (*F. major* vs. *P. kingtungense*), 2:2 (*F. major* vs. *R. pubescens*), 1:2 (*F. major* vs. *C. yunnanensis*), 1:1 (*F. major* vs. *Du. zibethinus*), 1:1 (*O. pyramidale* vs. *B. ceiba*), 1:1 (*O. pyramidale* vs. *F. major*) and 1:1 (*O. pyramidale* vs. *Du. zibethinus*). For more details of inter-paleopolyploid orthologous synteny, see also Supplementary Figs. 12–14. Source data are provided as a Source Data file.

*trichospermus*, and *P. kingtungense* (Fig. 2D, Supplementary Fig. 12A–C), indicating a shared tetraploidy event (herein referred to as Mal-β) (Fig. 3). Conversely, the paleotetraploid genome of *R. pubescens* showed a 2:2 synteny pattern with the three paleotetraploid genomes (Fig. 2D, Supplementary Fig. 12G–I), suggesting an independent tetraploidy event (Hel-β; Fig. 3). The genome of *C. yunnanensis* displayed clear 2:1 orthologous synteny patterns relative to the three paleotetraploid genomes (Fig. 2D, Supplementary Fig. 12D–F), indicating a shared Mal-β tetraploidy event followed by an additional lineage-specific tetraploidy event (Til-α; Fig. 3). Similarly, the pairwise comparisons of paleodecaploid genomes of *O. pyramidale*, *B. ceiba*, and *G. raimondii* showed 1:1 orthologous synteny patterns to each other (Fig. 2D, Supplementary Fig. 13A–C), supporting a shared decaploidy event (Mal-α; Fig. 3). In *Du. zibethinus*, two of three subgenomes showed a 1:1 orthology with the subgenomes of the three paleotetraploid species (*F. major*, *D. trichospermus* and *P. kingtungense*) (Fig. 2D, Supplementary Fig. 13D–F), implying that a related tetraploid ancestor contributed to the paleohexaploid origin of the *Du. zibethinus* genome (Dur-α; Fig. 3). Likewise, two of five subgenomes in the three paleodecaploid species (*O. pyramidale*, *B. ceiba* and *G. raimondii*) showed 1:1 orthology with the three paleotetraploid genomes (Fig. 2D, Supplementary Fig. 14A–I), and three of five subgenomes exhibited 1:1 orthology with the subgenomes of *Du. zibethinus* (Fig. 2D, Supplementary Fig. 13G–I), suggesting the hypothesis that the paleodecaploids potentially originated stepwise from the paleotetraploid and paleohexaploid genomes or their close relatives (Fig. 3).

We then employed phylogeny-based evidence to further elucidate these complex polyploidy processes. Using a methodology combining multiple lines of evidence (see Methods for details), we assigned subgenomes for simplicity: the two subgenomes of paleotetraploids (*F. major*, *D. trichospermus* and *P. kingtungense*) were denoted as A and B, the three subgenomes of *Du. zibethinus* as A, B and C, and the five subgenomes of the three paleodecaploids as A–E (Figs. 3 and 4, Supplementary Figs. 15–19). The two A/B subgenomes of *C. yunnanensis* were designated as A1 and A2, and B1 and B2, respectively, while the two subgenomes of *R. pubescens* were denoted as D1 and D2 (Supplementary Fig. 15). Chromosome- and subgenome-scale phylogenies (Supplementary Figs. 17–19) supported the conclusions drawn from orthologous synteny patterns and provided additional insight into the intricate relationships among subgenomes. Specifically, we found that all A subgenomes consistently formed a monophyletic group, as did the B and C subgenomes (Fig. 4, Supplementary Figs. 17–19). Both the A and B subgenomes in *Du. zibethinus* were sister to those of *F. major*, suggesting that the ancestor of the latter species or its close relative was the tetraploid progenitor of the paleohexaploid *Du. zibethinus* (Fig. 3). However, although the C subgenome of the paleodecaploids (*O. pyramidale*, *B. ceiba* and *G. raimondii*) was sister to the C subgenome of *Du. zibethinus*, the A and B subgenomes of the

paleodecaploids showed inconsistent sister relationships with *Du. zibethinus* or other A/B genomes (Supplementary Figs. 17–19). This evidence does not support the hypothesis that the paleohexaploid or its sisters were the hexaploid progenitor of the paleodecaploids[38]. Instead, the allodecaploid genome likely formed from another ancient hexaploid that arose independently through hybridization between an AABB tetraploid and a CC diploid closely related to the C progenitor of *Du. zibethinus* (Fig. 3). The two rounds of paleotetraploidy in *C. yunnanensis* (Fig. 3) were corroborated by the subgenome phylogenies (Supplementary Figs. 17–19) and distinct inter-subgenomic *K*s peaks (Supplementary Fig. 20A). However, similar patterns with differential inter-subgenomic *K*s peaks were not observed in *Du. zibethinus*, *O. pyramidale*, *B. ceiba* and *G. raimondii* (Supplementary Fig. 20B–E), likely due to the rapid divergence of their diploid progenitors.

Reticulate allopolyploidy generating at least two polyploids has been documented in lineages like wheats[50], oats[51], crucifers[52], *Prospero autumnale*[53], poppies[4] and bamboos[8]. However, none of these have as much complexity as the genomes of the Malvaceae (Fig. 3). The process of complex reticulate allopolyploidization has produced at least five ancient polyploid lineages with different ploidies, entailing one or two polyploidization events (i.e., Mal-β, Mal-β + Dur-α, Mal-β + Til-α, Mal-β + Mal-α, and Hel-β; Fig. 3). Thus, polyploid speciation certainly played an important role in the early divergence of the Malvadendrina clade.

## Origin and diversification of the ancient progenitors within the Malvaceae
To further elucidate these complex evolutionary histories, we reconstructed subgenome-aware phylogenetic trees by using coalescent, concatenation, and site-based methods, and congruent relationships were observed among the backbone clades (Supplementary Figs. 18 and 19). Coalescent and concatenation methods also revealed multiple rounds of species radiation, punctuated by the tetraploidization (Mal-β) and decaploidization (Mal-α) events (Supplementary Figs. 18 and 21; Supplementary Note 3). However, none of these methods yielded a consistent resolution between the A and B subgenome lineages across multiple species (Supplementary Figs. 18 and 19). The pronounced gene tree discordance (Supplementary Fig. 21) suggested that the evolutionary trajectory of diploid subgenomes may deviate from a strictly bifurcating model, necessitating network-based approaches to capture gene flow. Notably, split networks revealed strikingly similar topologies between A and B lineages (Fig. 4A), particularly when *O. pyramidale* represents *B. ceiba* and *G. raimondii* (Supplementary Fig. 22), potentially mitigating long-branch attraction artifacts[54]. Under the maximum parsimony criterion, these findings support a single origin for the ancient AABB tetraploids (Fig. 3) over multiple origins (Supplementary Fig. 23). This is further supported by the presence of a chromosomal rearrangement between the A and B subgenomes in all the AB-containing genomes (Fig. 3; for

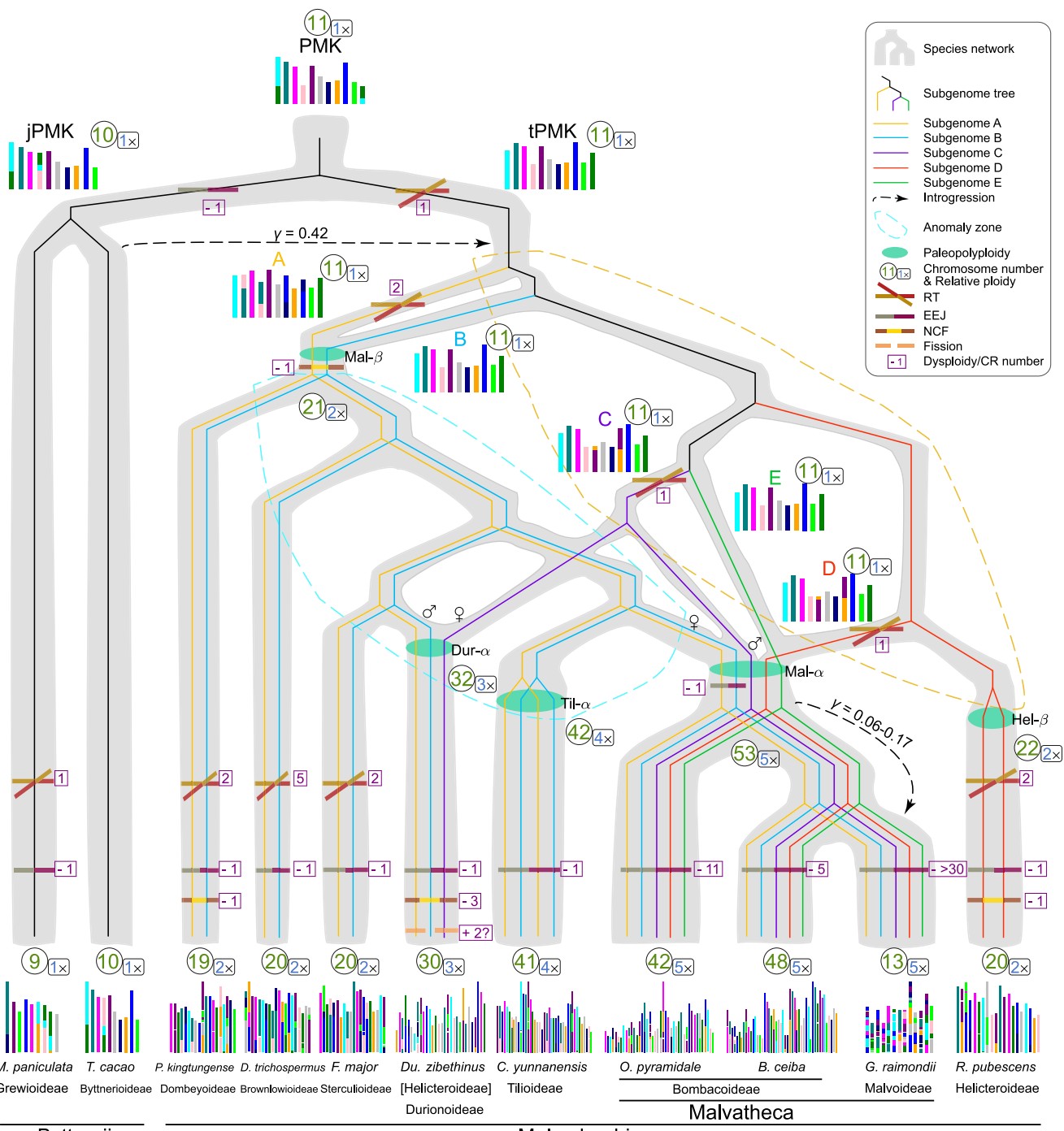

**Fig. 3 | Complex reticulate evolution of the Malvaceae genomes.** Paleopolyploidy events: Mal-β, an allotetraploidy event shared by the Malvadendrina clade, with the exception of *R. pubescens*; Mal-α, an allodecaploidy event shared by the Malvatheca clade. Dur-α, an allohexaploidy event specific to *Du. zibethinus*; Til-α, a tetraploidy (likely allotetraploidy) event specific to *C. yunnanensis*; Hel-β, a tetraploidy (likely allotetraploidy) event specific to *R. pubescens*. Large-scale interchromosomal rearrangements (CRs) between non-homologous chromosomes: RT reciprocal translocation, EEJ end-to-end joining, NCF nested chromosome fusion; an EEJ or NCF event reduces chromosome number by one (−1), while RT does not alter the chromosome number. PMK, proto-Malvaceae karyotype; tPMK and jPMK, two variants of PMK resulting from an RT and EEJ event, respectively. The circled numbers indicate either the haploid chromosome number (*n*) for ancestral genomes or the monoploid chromosome number (*x*) for modern genomes, and the numbers in the rounded box indicate the relative ploidy level (e.g., 2× indicates paleotetraploidy, and 5× indicates paleodecaploidy). The numbers in the violet, square box indicate the number of CRs, and '+' or '−' stands for ascending or descending dysploidy, respectively. Note that the unusual ascending dysploidy in the *Du. zibethinus* genome is most likely due to incorrect genome assembly. For more details supporting the evolutionary histories, see also Supplementary Figs. 11–15 (paleopolyploidy), 18–22 (subgenome phylogeny), 24–28 (introgression), and 33–38 (karyotype evolution and dysploidy). Source data are provided as a Source Data file.

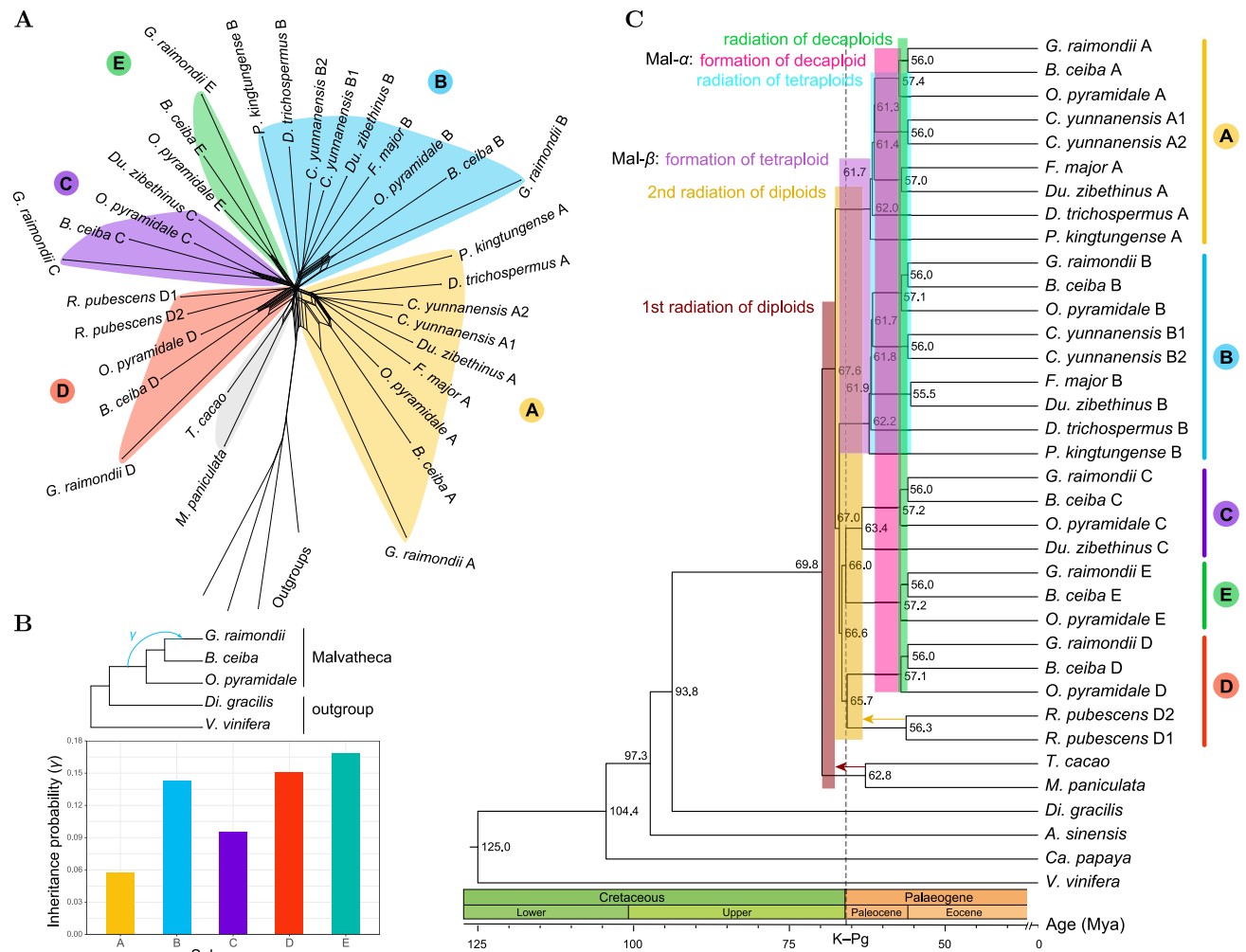

**Fig. 4 | Subgenome-aware phylogenetic analyses. A** The split network illustrates the intricate evolutionary relationships between the subgenomes (A–E) and highlights the complex, network-like evolution beyond simple, bifurcating evolutionary pathways. *Vitis vinifera, Carica papaya, Aquilaria sinensis* and *Dipterocarpus gracilis* served as outgroups. **B** Introgression event between three paleodecaploid species inferred from five subgenomes. *V. vinifera* and *D. gracilis* served as outgroups. Different subgenome data resulted in a convergent scenario (phylogenetic network), while the inferred inheritance probability (γ) values varied between the different subgenomes (bar chart) (see Supplementary Fig. 25 for more details). **C** Subgenome-aware phylogenetic relationships with estimated divergence times between subgenomes/species. Branch lengths and values in the nodes represent the estimated divergence time. Source data are provided as a Source Data file.

details, see below). The discordance between the A and B lineages in the subgenome trees could be attributed to biased ILS and/or hybridization/introgression during the radiation of the tetraploids or biased fractionation during the subsequent diploidization process. In addition, subgenome-aware phylogenetic analyses revealed that *Du. zibethinus* and *R. pubescens*, both members of Helicteroideae, had distinct paleopolyploidy origins during the early divergence of the Malvadendrina, and do not form a monophyletic group (Fig. 3, Supplementary Figs. 18 and 19). Consequently, we proposed a novel subfamily, Durionoideae R.-G. Zhang, subfam. nov., to accommodate *Du. zibethinus* and its closely related taxa in the tribe Durioneae[55] (see detailed taxonomic treatment in Supplementary Note 4 [page S4]).

Furthermore, explicit networks were constructed with internal nodes indicating speciation or hybridization/introgression events. During the initial radiation of the ancient diploids, the network revealed an edge with an inheritance probability (γ) of 0.42, indicating introgression from the progenitor of *T. cacao* to the progenitor of the Malvadendrina clade (Supplementary Fig. 24, Fig. 3). In the radiation of ancient decaploids, five networks determined by the five subgenomic datasets (A–E) consistently suggested introgression from a ghost lineage into the ancestral lineage of *G. raimondii*, with γ values ranging from 0.06 (subgenome A) to 0.17 (subgenome E) (Figs. 4B and 3, Supplementary Figs. 25 and 26). No reliable network showed a goodness-of-fit to gene tree concordance patterns for the second radiation of ancient diploids and the radiation of AABB paleotetraploids (Supplementary Figs. 27 and 28), hinting at anomaly zones[56] (Fig. 3), as evidenced by the prevalence and high degree of both ILS and introgression signals (Supplementary Fig. 21).

Reconciling the tree topologies of the A and B lineages with phylogenetic trees and networks, we estimated the divergence times among subgenomes/species and the time frame of paleopolyploidy events (Fig. 4C). The crown age of the Malvaceae is estimated at approximately 70 million years ago (Mya), with the radiation of multiple diploid ancestors of Malvadendrina paleopolyploids occurring around 68–63 Mya (Fig. 4C). The AABB paleotetraploid (Mal-β) likely formed between 67 and 62 Mya, while the AABBCCDDEE paleodecaploid (Mal-α) emerged between 61 and 57 Mya (Fig. 4C). These Malvadendrina paleopolyploidy events potentially coincided with the period of global climate change at the Cretaceous–Paleogene (K–Pg) boundary (-66 Mya). However, due to dating uncertainties, it is unclear whether the paleopolyploidy events predated or postdated the K–Pg

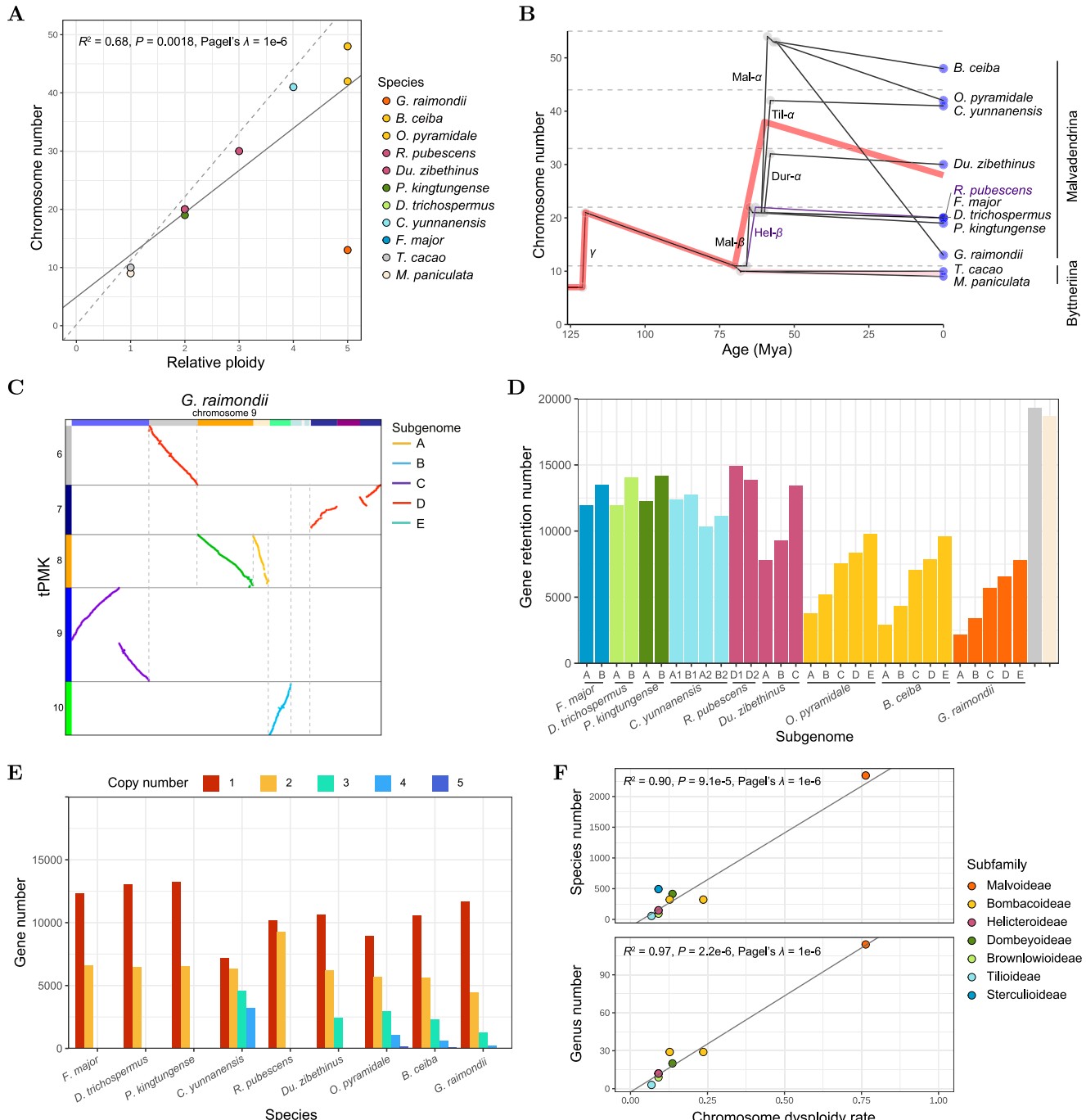

**Fig. 5 | Patterns and implications of post-polyploid diploidization (PPD) in Malvaceae. A** Correlation between chromosome number and relative ploidy ($p$) in the analyzed species. The dashed line indicates the expected chromosome number ($11 \times p$) in the absence of dysploidy. All genomes analyzed, with the exception of the cotton genome, have chromosome numbers close to the chromosome numbers of their polyploid ancestors. **B** Polyploidy–dysploidy waves drive chromosome number evolution in Malvaceae. The black lines represent the chromosome number changes across a reconciled species tree. The red line represents the average of chromosome numbers in the Malvadendrina clade. **C** A typical case of EEJ fusions identified in the cotton genome. The dots were colored according to subgenome assignments. At least six

ancestral chromosomes formed chromosome 9 in *G. raimondii* ($n = 13$) during diploidization of the paleodecaploid genome ($n = 53$). See more EEJ fusions in Supplementary Fig. 38. **D** Biased fractionation patterns indicate progressive subgenome fractionation from A to E (A > B > C > D > E). **E** Retained copy number of multiplicated genes in each paleopolyploid. **F** Correlation between the extent of PPD and evolutionary success expressed as genus/species richness in the seven Malvadendrina subfamilies. Chromosome dysploidy rate quantifies the extent of PPD at the chromosome level. The numbers of genera and species were adopted from the World Flora Online (WFO) database. A *t* test was used in the correlation analyses (**A** and **F**) under a phylogenetic generalized least squares framework. Source data are provided as a Source Data file.

boundary; an alternative estimate places the paleotetraploid formation at approximately 77–69 Mya, i.e., before the K–Pg boundary (Supplementary Fig. 29). Nonetheless, integration of molecular dating and fossil evidence (Fig. 4C, Supplementary Figs. 29 and 30) confirms

that paleopolyploidy events occurred around the K–Pg boundary, with all Malvaceae subfamilies diverging before the upper boundary of the late Paleocene (-56 Mya), prior to the Paleocene–Eocene Thermal Maximum (-55 Mya).

## Descending dysploidy dominated chromosome-level PPD in the Malvaceae

Chromosome number variation across Malvaceae species typically reflects the ancestral polyploidization events (Fig. 5A, Supplementary Fig. 31). Specifically, the paleotetraploids *F. major*, *D. trichospermus*, *P. kingtungense* and *R. pubescens* ($x = 19-20$; $p = 2$) exhibit roughly twice the chromosome count of the paleodiploid Byttneriina genomes ($x = 9-10$; $p = 1$). Similarly, the paleohexaploid *Du. zibethinus* shows a threefold increase in chromosome number ($x = 30$; $p = 3$), *C. yunnanensis* a fourfold increase ($x = 41$; $p = 4$) and the paleodecaploids *O. pyramidale* and *B. ceiba* a fivefold increase ($x = 42$ and $48$; $p = 5$). However, the paleodecaploid cotton species *G. raimondii* has an atypically low chromosome number ($x = 13$; $p = 5$), contrasting sharply with its sister Bombacoideae species ($x = 42$ and $48$; Fig. 5A). In contrast to the positive correlation between paleoploidy levels and chromosome numbers ($R^2 = 0.68$, $P = 0.0018$, Pagel's $\lambda = 1e-6$; Fig. 5A), genome sizes do not correlate with paleoploidy levels ($P > 0.1$, Pagel's $\lambda = 1e-6$; Supplementary Fig. 31B). Instead, genome size is highly correlated ($R^2 > 0.95$, $P < 0.001$, Pagel's $\lambda = 1e-6$) with the content of TEs and LTR-RTs which expanded over the past few million years (Supplementary Figs. 31C, D and 32).

To trace chromosome evolution through recurrent paleopolyploidization and diploidization processes, we reconstructed ancestral karyotypes[57,58], including the proto-Malvaceae karyotype (PMK) and its more recent, translocation-carrying variant, tPMK, representing the proto-Malvadendrina karyotype (Supplementary Figs. 15 and 33). Both PMK and tPMK comprised 11 proto-chromosomes, the number that is consistent with recent studies[38,49]. PMK and tPMK differed by a reciprocal translocation (RT) post-dating the divergence of the Byttneriina and Malvadendrina clades (Fig. 3, Supplementary Fig. 34). In the Byttneriina clade, an end-to-end joining (EEJ) between chromosomes PMK4 and PMK11 mediated descending dysploidy from $n = 11$ to $n = 10$, with a further EEJ in *M. paniculata* yielding $x = 9$ (Supplementary Fig. 34).

The established tPMK allowed us to reconstruct chromosome evolution in the context of paleopolyploidizations in the Malvadendrina (Figs. S35–S39). Each proto-chromosome of tPMK was corroborated by its presence in the extant genomes of at least four of the seven Malvadendrina subfamilies (Supplementary Figs. 15 and 33). The tPMK genome differentiated into A-genome and B-genome diploids with two RTs in the A-genome ancestor (Supplementary Fig. 35). Hybridization between A and B diploids (Mal-β) doubled the chromosome number to $n = 22$ (Fig. 5B), followed by a nested chromosome fusion (NCF) in the AB paleotetraploid (Supplementary Fig. 36) reducing the chromosome number to $n = 21$ (Fig. 3). As the NCF event involved chromosomes of both subgenomes, it should have occurred after the formation of the allotetraploid genome (Fig. 3). Shared RTs were also identified within the C and D subgenomes (one RT in the C subgenome and another in the D subgenome), respectively (Fig. 3, Supplementary Fig. 37). Similar to the B subgenome, the diploid E genome retained the ancestral tPMK-like structure (Fig. 3, Supplementary Fig. 15). After the Mal-α paleodecaploidy event (AABBCCD-DEE), the chromosome number increased to $n = 54$ (Figs. 3 and 5B). An EEJ between subgenomes A and B, shared by all Malvatheca decaploid genomes (Supplementary Fig. 38A, B), indicates that the reduction of chromosome number from $n = 54$ to $n = 53$ occurred prior to the diversification of the Malvatheca species (Fig. 3). Post-polyploid chromosomal diploidization was predominantly mediated by EEJs in the Malvatheca clade (Supplementary Fig. 38C, D), resulting in modern chromosome numbers of $x = 48$, $42$ and $13$ in the three species analyzed (Figs. 3 and 5B). Notably, over 30 lineage-specific EEJ events occurred during the evolution of the paleodecaploid *G. raimondii* genome (Fig. 5C, Supplementary Fig. 38C, D), resulting in a drastic fourfold reduction in chromosome number from 53 to 13 (Figs. 3 and 5B). These extensive descending dysploidy events should

occur between the emergence of the Malvoideae (~56 Mya) and the diversification of the tribe Gossypieae (and *Gossypium*) (~10–20 Mya), with $x = 13$ being the most common chromosome number[29,59-61]. In contrast, only 5–11 lineage-specific EEJ events were inferred for the two Bombacoideae genomes (Fig. 3, Supplementary Fig. 38C, D). Lineage-specific EEJs also occurred in the other six paleopolyploid genomes analyzed, while NCFs occurred in three of these genomes (Fig. 3, Supplementary Fig. 39). These chromosomal rearrangements led to a moderate reduction in chromosome number (−1 to −4 chromosomes) in these clades (Supplementary Fig. 39). An unusual increase in chromosome number, attributable to chromosome fission, was observed in the *Du. zibethinus* genome (Supplementary Fig. 39). However, these two fissions are most likely due to assembly artifacts, as the number of assembled pseudochromosomes ($x = 30$) is higher than the reported chromosome count ($x = 28$)[36].

In summary, the predominant trend in PPD in the Malvaceae was a reduction in chromosome number (descending dysploidy) mediated by EEJs (Figs. 3 and 5B). Considering the known chromosome numbers of species not sequenced, descending dysploidy events in Brownlowioideae, Dombeyoideae, Sterculioideae and Tilioideae may have occurred in the stem branch of these subfamilies (Supplementary Fig. 40). As polyploidization, dysploidy, or other large-scale chromosomal rearrangements can lead to reproductive isolation and speciation, chromosomal variation is likely to be a potentially important driver of speciation and cladogenesis in the Malvaceae. Overall, our results emphasize the importance of polyploidy–dysploidy waves (Fig. 5B) for the evolution of plant genomes.

We further quantified the rate of descending dysploidy by dividing the diploidized chromosome number by the expected chromosome number of the paleopolyploid ancestor (see Methods). The descending dysploidy rates for the woody Malvadendrina species (0.07–0.24; Supplementary Table 8), with the exception of *G. raimondii* (0.76), are in a similar range as dysploidy rates (range 0–0.5) in trees or shrubs with a comparable age of WGDs (~50–80 Mya)[62-65]. Moreover, our results show that dysploidy rates are significantly positively correlated with nucleotide substitution rates ($R^2 = 0.45$, $P = 0.025$, Pagel's $\lambda = 1e-6$; Supplementary Fig. 41A), whereas maximum plant height is negatively correlated with nucleotide substitution rates ($R^2 = 0.56$, $P = 0.0079$, Pagel's $\lambda = 1e-6$; Supplementary Fig. 41B), consistent with slower molecular evolution in taller plants[66]. Shorter plants (e.g., *G. raimondii*) may have faster long-term rates of meiosis and faster mitotic divisions in the apical meristems, leading to the accumulation of more heritable mutations[66], including dysploidal chromosomal rearrangements. It should be informative to compare the $x = 13$ cotton genome with more diploidized Malvoideae genomes with 5–12 chromosome pairs (e.g., $n = 5$ in *Sphaeralcea*).

## Biased fractionation and fates of duplicated genes in the genic PPD processes in Malvaceae

In addition to descending dysploidy, gene fractionation, or gene loss, marks another key feature of the PPD process. Analysis of AB subgenomes in Malvadendrina paleopolyploids revealed a biased fractionation pattern, with the A subgenome exhibiting a lower gene retention rate than the B subgenome (Wilcoxon rank-sum test, $P < 0.0001$; Fig. 5D, Supplementary Fig. 42). This consistent bias (A < B) likely predated the divergence of ancient tetraploids. Notably, species with AABBCC (*Du. zibethinus*) and AABBCCDDEE (*O. pyramidale*, *B. ceiba* and *G. raimondii*) genome composition retained significantly fewer genes in their A/B subgenomes compared to the AABB tetraploid species (*F. major*, *D. trichospermus* and *P. kingtungense*) (Wilcoxon rank-sum test, $P < 0.0001$; Fig. 5D, Supplementary Fig. 42), maintaining the A < B trend. In the allohexaploid AABBCC genome (*Du. zibethinus*), the C subgenome showed a significantly higher retention rate than A or B (A < B < C; Wilcoxon rank-sum test, $P < 0.0001$) (Fig. 5D, Supplementary Fig. 42). This observation aligns with the two-step model of

hexaploid genome formation, where the most recently added sub-genome retains the most genes[67]. The allodecaploid AABBCCDDEE species (*O. pyramidale*, *B. ceiba* and *G. raimondii*) also showed a progressive retention gradient from A to E (A < B < C < D < E) (Fig. 5D, Supplementary Fig. 42), implying a four-step pathway in which subgenomes C, D and E merged successively with the ancestral tetraploid AABB genome (Supplementary Fig. 43). Specifically, the evolutionary trajectory of the ancient decaploid can be envisioned as a sequence of hybridization events with unreduced gametes[68], evolving from a tetraploid (AABB) to a hexaploid (AABBCC), then to an octoploid (AABBCCDD), and finally to a decaploid (AABBCCDDEE) (Supplementary Fig. 43, Fig. 3). Nevertheless, this hypothesis could be strengthened by the inclusion of genome analysis of intermediate hexaploid and octoploid genomes, providing clues for further investigation. In addition, the highly retained homoeologs also showed biased expression in most comparisons (Wilcoxon rank-sum test, $P < 0.05$), with a similar trend of gene retention (i.e., A < B < C < D < E) (Supplementary Fig. 44A). However, this trend appeared to be disrupted in the *G. raimondii* genome, which exhibited an A/B < C/D/E trend (Supplementary Fig. 44A). This anomaly may be caused by the extreme PPD in this cotton genome. In addition, there is also biased fractionation between the subgenomes generated by Til-α and Hel-β (Wilcoxon rank-sum test, $P < 0.0001$; Supplementary Fig. 42A), implying that these WGD events could also be allopolyploidizations[69].

After long-term fractionation spanning ~60 million years, most duplicated gene groups were retained as single copies in all paleopolyploids (Fig. 5E). The retention of multiple copies was drastically reduced in the allohexaploid and allodecaploid genomes (Fig. 5E). In particular, only 9% of genes with three to five copies were retained in the cotton genome. For example, only 20 five-copy genes were retained, compared to 86–195 in the Bombacoideae genomes (Fig. 5E). For the highly retained duplicates (at least 80% copies retained), regulatory genes encoding transcription factors were over-represented (Supplementary Fig. 45). This observation is likely consistent with the dosage-balance hypothesis[70,71] that predicts enrichment of dosage-dependent genes, including regulatory genes.

## The importance of polyploidy for survival and PPD for success

Based on our findings and previous phylogenetic studies[29], all species within the Malvadendrina clade are likely to have a paleopolyploid origin, with their five paleodiploid progenitor lineages (AA–EE) presumably extinct. The formation and diversification of these paleopolyploid genomes around the K–Pg boundary and predating the Paleocene–Eocene Thermal Maximum (Fig. 4C) support the link between WGDs and survival of flowering plants during the environmental catastrophes[9,72]. We speculate that the establishment of the polyploid ancestors may have been facilitated by the vacant niches opened up by the K–Pg extinction. However, except for the Malvoideae, most paleopolyploid Malvadendrina subfamilies contain fewer genera and species than the diploid Byttneriina subfamilies, and no correlation was evident between the relative ploidy level and species/genus richness ($P > 0.3$, Pagel's $\lambda = 1e-6$; Supplementary Fig. 46A, B), suggesting that polyploidy does not guarantee evolutionary success defined as genus or species richness of a given subfamily.

Given that all subfamilies within the Malvadendrina are paleopolyploids formed around 60 Mya, why do some appear to be more successful than others? The Malvoideae subfamily represented by *Gossypium* is characterized by the most progressed diploidization, as evidenced by the lowest chromosome number and minimal retention of syntenic genes compared to other paleopolyploids (Fig. 5A, Supplementary Fig. 31A). While the Malvoideae can be considered the most successful subfamily in the family, as evidenced by the highest number of genera and species (~110 genera and ~2300 species, the World Flora Online database), the sister Bombacoideae (~30 genera and ~320 species) contains four times fewer genera and seven times

fewer species than the Malvoideae. The Malvoideae have a mean chromosome number ($n = $~20) similar to that of the paleotetraploid Sterculioideae, but much lower than that of the Bombacoideae ($n = $~50)[61]. As both Malvatheca subfamilies share the ancestral decaploid genome (Mal-α), a more extensive PPD in the Malvoideae could be important for the evolutionary success and diversification of this subfamily. We further tested this hypothesis using the whole dataset from Malvadendrina paleopolyploids. Our results showed strong positive correlation between the extent of chromosomal diploidization (i.e. dysploidy rate) and the taxonomic richness of the paleopolyploid subfamilies ($R^2 = 0.97$, $P < 1e-5$, Pagel's $\lambda = 1e-6$ for genus richness, and $R^2 = 0.90$, $P < 1e-4$, Pagel's $\lambda = 1e-6$ for species richness; Fig. 5F). At the gene level, the association between the extent of PPD (i.e., gene loss rate) and taxonomic richness was also supported at the genus level ($R^2 = 0.44$, $P = 0.053$, Pagel's $\lambda = 1e-6$) but little supported at the species level ($R^2 = 0.24$, $P = 0.18$, Pagel's $\lambda = 1e-6$) (Supplementary Fig. 46C, D). These findings support the importance of PPD for diversification. Based on the fact that the correlation coefficients are much higher at the chromosome level than at the gene level, however, we speculate that chromosomal diploidization (descending dysploidy) played a more important role in the diversification of paleopolyploid clades than genic diploidization (gene loss).

The hypothesis that PPD leads to success can reconcile the disparate perspectives on the relative success of polyploids versus diploids[13,18,19,73]. Polyploidy is known for its dual advantages and disadvantages[74]. The latter, such as the challenges posed by nuclear enlargement and meiotic abnormalities due to unstable chromosome behavior[74], can potentially be mitigated by diploidization of the genome[75]. Thus, through genic (e.g., loss or neofunctionalization of duplicated genes) and chromosomal (intra- and inter-chromosomal rearrangements) diploidization[75], the advantages of polyploidy can be preserved and enhanced. Conversely, unresolved disadvantages such as hybrid sterility and small initial population size could result in extinction and evolutionary 'dead ends'[18]. Therefore, a PPD process that takes millions of years could explain the observed discrepancy that recently formed polyploids have not yet achieved success, whereas many ancient polyploids have become successful[18,19]. This also explains the observed lag time in diversification following WGDs (i.e., the WGD Radiation Lag-Time Model) in flowering plants[13,14]. Indeed, a lag time (~10 million years) between the paleodecaploidy event (Mal-α) and the major diversification of the Malvatheca clade (i.e., radiation of Malvoideae) further exemplifies this pattern[29,76].

If our hypothesis holds true, another question is why chromosome reduction (descending dysploidy) seems to be more important for evolutionary success than gene loss (Fig. 5F, Supplementary Fig. 46C, D). First, the cytogenetic disadvantages of polyploidy, such as the irregular pairing of homeologous chromosomes, often lead to reduced fertility[77], probably exerting more deleterious effects than those associated with duplicated gene sequences. Therefore, chromosomal diploidization, including descending dysploidy, may be more effective than genic diploidization (e.g., gene fractionation). Second, dysploidy, directly leading to karyotypic diversity[78], can enforce or reinforce speciation and diversification[22], eventually leading to increased genetic and taxonomic richness. For example, among about 40 dysploidy events leading to the extant cotton genome (Figs. 3 and 5B), each event may increase barriers to gene flow and contribute to speciation of other lineages[79]. Recently, karyotypic diversity (number of distinct chromosome numbers) formed by polyploidy and dysploidy has been found to be positively correlated with species richness and diversification rates at both genus and family levels in flowering plants[80]. In contrast to dysploidy, linking post-polyploid gene loss to speciation events is less straightforward. For example, the *G. raimondii* genome has experienced extensive gene losses (~70,000 duplicated genes lost; Supplementary Fig. 31A), most of which were likely neutral and may not contribute to speciation[81].

The dysploidy rate or PPD rate need not be constant, but may gradually decrease over time (Supplementary Fig. 47), which could potentially explain why the correlation coefficients are higher at the genus level than at the species level (Fig. 5F, Supplementary Fig. 46C, D). In the early phase after allopolyploidization, genetic similarity between homoeologous chromosomes may increase karyotypic instability and thus promote chromosomal rearrangements, including descending dysploidy[82,83]. Over tens of millions of years, the probability of descending dysploidy likely decreases as non-homologous chromosomes are genetically more differentiated and their numbers are fewer[22,79]. For example, most species of the tribe Gossypieae, including *Gossypium*, have not undergone dysploidy changes in the last 20 million years[59–61]. Consequently, the impact of PPD may persist at the genus level but later decrease at the species level (Supplementary Fig. 47A), likely resulting in a lower correlation between the extent of PPD and species richness (Fig. 5F, Supplementary Fig. 46C, D). Future studies with expanded taxon sampling could refine the temporal dynamics of dysploidy and diversification rates to further test this hypothesis. We might also expect a shorter lag time between dysploidy and diversification compared to that between polyploidy and diversification (Supplementary Fig. 47B).

Although here we have provided insights into the importance of polyploidy and PPD, we also acknowledge certain limitations to our ability to infer these past processes. One major limitation is the sampling of genomes (Supplementary Note 5 lists additional limitations). While our study includes genomes from all nine Malvaceae subfamilies, some subfamilies are represented by only one species, potentially limiting the capture of within-subfamily variation in dysploidy rates. However, almost all selected species have chromosome numbers close to their subfamily means or medians (Supplementary Fig. 48)[61], suggesting that the selected genomes have reasonable representativeness. Nonetheless, we expect that expanded taxon sampling in future studies may reveal more variations, particularly in subfamilies with high species diversity.

## Methods

### Genome sequencing

The plant samples were collected from two locations: *D. trichospermus* from the Xishuangbanna Tropical Botanical Garden in Mengla, and *P. kingtungense*, *C. yunnanensis*, and *R. pubescens* from the Kunming Botanical Garden in Kunming, both part of the Chinese Academy of Sciences. The genomic DNA was extracted using a modified CTAB method[84].

For whole genome sequencing, genomic DNA (1 µg) was fragmented into sizes of 200–400 bp using a Covaris sonicator (Brighton, UK). These fragments were used to prepare short-read libraries according to the manufacturer's protocols. Short-read sequencing was performed on the DNBSEQ-T7 platform (BGI Inc., Shenzhen, China) in paired-end 150 (PE150) mode. Long-read sequencing was facilitated by fragmenting the genomic DNA with the Megaruptor 3 shearing kit (Diagenode SA, Seraing, Belgium). Fragments smaller than 5 kb were selectively depleted using the AMPure PB Beads size selection kit. Pacbio SMRTbell Prep Kit 3.0 was used to prepare the sequencing libraries, which were then sequenced on the Pacbio Revio System (Pacbio, Menlo Park, CA, USA) in high-fidelity (HiFi) mode.

For Hi–C sequencing, young shoot leaf material was fixed in 2% formaldehyde in 1× PBS buffer, and the library was prepared as follows[85]. Chromatin was digested with *Mbo*I, tagged with biotin-14-dCTP and ligated. After ligation, the DNA was reverse cross-linked and purified, and then sheared to 200–600 bp fragments. These biotin-labeled chromatin fragments were enriched using streptavidin magnetic beads and, after A-tailing and adapter ligation, amplified by PCR (12–14 cycles) and sequenced on the DNBSEQ-T7 platform in PE150 mode.

Full-length transcriptomic data were obtained through Iso-seq. RNA was extracted from various tissues using the R6827 Plant RNA Kit (Omega Bio-Tek, Norcross, GA, USA) following the manufacturer's instructions. Full-length cDNA libraries were prepared using the cDNA-PCR Sequencing Kit (SQK-PCS109) and sequenced using the Pro-methION sequencer (Oxford Nanopore Technologies, Oxford, UK).

### Genome assembly

To guide genome assembly, $k$-mer ($k = 19$) profiles of PacBio HiFi reads were produced using KMC v3.1.1[86], allowing the estimation of genome size, heterozygosity rate and ploidy using GenomeScope 2.0 and Smudgeplot v0.2.3[87]. Both HiFi and Hi-C reads were then input into Hifiasm v0.19.5[88,89] to assemble haplotype-resolved contigs. Then Hi-C reads were mapped to these contigs using Juicer v1.5.6[90], and chromosome scaffolding was conducted using the 3D-DNA v180922[91] pipeline with parameters '--early-exit -m haploid -r 0'. Misassemblies and switch errors in the chromosomes were corrected manually using Juicebox v1.11.08[92], resulting in refined chromosome-scale scaffolds.

To enhance continuity, LR_Gapcloser[93] was employed to fill gaps in the chromosome assembly using HiFi reads (-s p -r 2 -g 500 -v 500 -a 0.25). HiFi reads were then re-mapped to the chromosome scaffolds. The mapped reads located around the chromosome ends were extracted and assembled into contigs using Hifiasm. Subsequently, these contigs were remapped to the scaffolds to extend the chromosome ends, facilitating the assembly of telomeric (TTTAGGG) repeats[94]. Chloroplast and mitochondrial genomes were assembled using GetOrganelle v1.7.5[95] and Bandage v0.9.0[96] with manual curation. To further refine the assembly, Nextpolish2 v0.1.0[97] was used to polish the assembly with both HiFi and short reads. Redundant haplotigs and rDNA fragments were removed using the Redundans[98] v0.13c pipeline (--identity 0.98 --overlap 0.8) with manual curation.

The overall assembly quality was assessed using KAT v2.4.2[39] against the HiFi reads, and BUSCO v5.3.2[40] against the embryophyta_odb10 database (1614 conserved genes).

### Genome annotation

The de novo identification of transposable elements using the EDTA (--sensitive 1) pipeline v1.9.9[99] generated a TE library. Subsequently, repetitive elements were identified with RepeatMasker v4.0.7[100].

To annotate the protein-coding genes, a data set comprising 314,962 non-redundant protein sequences from related plant species (*Theobroma cacao*[42], *Corchorus capsularis*[101], *Heritiera littoralis*[102], *Durio zibethinus*[36], *Bombax ceiba*[103], *Gossypium raimondii*[59], *Dipterocarpus turbinatus*[104], *Aquilaria sinensis*[105], *Arabidopsis thaliana*[106], *Carica papaya*[107] and *Vitis vinifera*[48]) served as protein evidence. The Iso-seq data were mapped to the genome using minimap2 v2.24[108], and the transcripts were assembled using StringTie v2.1.5[109]. The gene structure was then annotated by PASA v2.4.1[110]. The ab initio prediction parameters were trained and optimized for five rounds in AUGUSTUS[41] v3.4.0 using the full-length gene set. Gene annotation was conducted using the MAKER2 v2.31.9 pipeline[111], integrating ab initio predictions with transcript and protein evidence. EvidenceModeler (EVM) v1.1.1[112] was then applied to synthesize the PASA and MAKER2 results and generate integrated consensus annotations. Additionally, TEsorter v1.4.1[113] identified TE protein domains within the genome (-genome -db rexdb -cov 30 -eval 1e-5 -prob 0.9), which were subsequently masked during EVM processing. The annotations of EVM were enhanced by including UTR sequences and alternative splicing by using PASA. All annotations considered too short (less than 50 amino acids) or missing start or stop codons, as well as those containing internal stop codons or ambiguous bases, were systematically excluded. The quality and completeness of the final gene annotation were assessed using BUSCO v5.3.2[40]. Transfer RNA (tRNA) and ribosomal RNA (rRNA) genes were identified using tRNAScan-SE v1.3.1[114] and Barrnap (https://github.com/

tseemann/barrnap) v0.9, respectively. RfamScan v14.2[115] was used to identify other types of non-coding RNA (ncRNA).

Three analytical approaches were used to elucidate the functions of protein-coding genes within the genome. The first method involved the use of eggNOG-mapper v2.0.0[116], which searched for homologous genes in the eggNOG database and enabled the annotation of genes with the Gene Ontology (GO) and the Kyoto Encyclopedia of Genes and Genomes (KEGG). In the second method, DIAMOND v0.9.24[117] was used to align protein-coding genes against several public protein databases, including Swiss-Prot, TrEMBL, NR and TAIR10 (--evalue 1e-5 --max-target-seqs 5). Finally, InterProScan v5.27[118] was used to annotate protein domains and motifs by matching multiple databases such as PRINTS, Pfam, SMART, PANTHER, and CDD of the InterPro consortium.

## Genomic data standardization

Representative genomes for other subfamilies and outgroups come from public repositories or were acquired through personal communications (Supplementary Tables 6 and 7). In the case of haplotype-resolved assemblies, one set of homologous chromosomes was extracted as the primary monoploid nuclear assembly from the multiple sets of homologous chromosomes. This selection was based on the highest number of orthologous syntenic genes to the cacao genome (see Methods detailed below). For genes represented by multiple transcripts, the isoform encoding the longest protein sequence was selected for analysis.

## Identification of orthologous synteny

An all-versus-all comparison of protein sequences was performed using DIAMOND v0.9.24[117]. Orthologous relationships were determined using OrthoFinder v2.3.1, with the parameter set to '-M msa'[119]. Syntenic blocks between and within genomes were identified using the '-icl' option of WGDI[57] v0.6.2, adhering to default parameters. The non-synonymous substitution rate ($K$a) and synonymous substitution rate ($K$s) for pairs of homologous genes were estimated using the '-ks' option of WGDI. Orthologous syntenic blocks were identified using the Orthology Index by integrating the orthology and synteny identified in the above steps[46].

## Reconstruction of the species tree

The species tree based on nuclear DNA was reconstructed using the SOI pipeline[46]. A total of 16,370 multi-copy gene trees derived using the default parameters of the SOI pipeline were used to infer a species tree using the coalescent-based method ASTRAL-Pro v1.15.2.3[120]. Additionally, a concatenated alignment of 1904 single-copy genes (SOI parameter: 'phylo -sc -mm 0.2 -concat') was compiled to reconstruct a phylogenetic tree using IQ-TREE v2.2.0.3[121], incorporating 1000 bootstrap replicates.

For plastid and mitochondrial genomes, PCGs were annotated using the OGAP pipeline (https://github.com/zhangrengang/ogap), aligned using MAFFT v7.481[122] and trimmed using trimAl v1.2[123] with the parameter '-automated1'. Finally, the aligned PCGs were concatenated, and maximum-likelihood trees were reconstructed with IQ-TREE v2.2.0.3[121] using 1000 bootstrap replicates.

## Reconstruction of ancestral karyotypes

Following the methodology of Sun et al.[57,58], we reconstructed ancestral karyotypes by identifying chromosome-scale orthologous synteny retained between Byttneriina and Malvadendrina genomes, as evidenced by dot-plot comparisons (see below); intra-chromosomal rearrangements, such as inversions and insertions/deletions, were ignored. The dot plots showed that eight (chr2, 3, 5, 6, 7, 8, 9 and 10) out of ten cocao chromosomes had chromosome-scale synteny with at least two other species (Supplementary Fig. 11). This allowed us to directly infer these chromosomes as ancient chromosomes of their

most recent common ancestor (MRCA). The remaining two cocao chromosomes (chr1 and 4) were formed by end-to-end joining (EEJ) and reciprocal translocation (RT) events when compared with other Malvadendrina genomes (Supplementary Fig. 11). Analysis in comparison with the grape genome suggests that the EEJ event is specific to Byttneriina genomes, while the RT is unique to Malvadendrina genomes (Supplementary Fig. 16). This hypothesis was further corroborated by comparisons with the genomes of *Buxus austroyunnanensis* and *Cercidiphyllum japonicum* (Supplementary Fig. 16), which underwent the least number of inter-chromosomal rearrangements as compared with the ancestral eudicot karyotype (AEK) and post-γ AEK, respectively[124]. The segregating states of proto-chromosomes 4 (PMK4) and 11 (PMK11) of the proto-Malvaceae karyotype (PMK) in the outgroup genomes (Supplementary Fig. 16) also supported the ancestral character of the PMK4/PMK11 fusion in the Byttneriina clade, in comparison with an alternative scenario—fission of a proto-chromosome PMK4 + PMK11 in the Malvadendrina clade. Consequently, we used the cocao chromosomes as a backbone to reconstruct the PMK by splitting chr4 and the tPMK by accounting for the RT observed between chromosomes 1 and 4. The identified inversions specific to the cocao genome were manually reversed. To refine and improve the tPMK, we pruned non-syntenic and tandemly repeated genes from the backbone and included genes that showed synteny to other genomes analyzed. The final tPMK comprises 27,564 anchor genes. Syntenic blocks between the tPMK and extant genomes were identified using the '-icl' option of WGDI, with orthologous synteny further identified using SOI (parameter 'filter -c 0.4')[46]. Finally, we generated a hierarchical gene list (one subgenome/species per column) using WGDI (with options '-bi', '-pc', '-a') using the tPMK as a reference. The karyotypes of the extant genomes were mapped to the tPMK also using the '-km' option of WGDI.

## Polyploidy inference and subgenome phasing

Initial polyploidy inference was conducted by examining orthologous synteny patterns[46] with reference genomes and $K$s peaks of paralogous syntenic genes. By comparison with the grape or cacao genomes, which have not undergone lineage-specific polyploidy events since the shared paleohexaploidy event, relative ploidy ($p$) to the reference genomes was determined from the orthologous syntenic depth ratio observed in dot plots (Supplementary Fig. 11). For example, the *F. major* genome showed a clear orthologous syntenic depth ratio of 2:1 ($p = 2$) to the cacao/grape genome (Supplementary Fig. 11A), so it should undergo a lineage-specific tetraploidy event, displaying two-fold ploidy relative to the cacao/grape genome. The polyploidy event was further cross-validated by the $K$s distribution patterns of paralogous syntenic genes, as a $K$s peak is expected for a polyploidy event. This strategy was also applied to other genomes to infer paleopolyploidy.

We then placed the polyploidy events into the species tree based on orthology patterns and chromosome/subgenome-scale phylogeny. The former was based on the principle of Zhang et al.[46]. For example, *F. major* and *P. kingtungense* showed clear 1:1 orthology + 1:1 out-paralogy patterns (Supplementary Fig. 12C), so they should share the tetraploidy event. In contrast, *F. major* and *R. pubescens* displayed orthology of nearly 2:2 (Supplementary Fig. 12G), so they each likely experienced an independent tetraploidy event. However, the relationships between the subgenomes of multiple species remained unclear, so phylogenetic analysis was further performed with phased subgenomes.

We used *F. major* to represent *D. trichospermus* and *P. kingtungense*, and *O. pyramidale* to represent *B. ceiba* and *G. raimondii*, as these two groups clearly share polyploidy history based on orthology patterns (Supplementary Figs. 12A–C and 13A–C, respectively). We then assigned the subgenomes based on orthology, synteny and phylogenetic evidence, following the guideline of Zhang et al.[5] with

minor modifications. SubPhaser failed to identify subgenome-specific repetitive *k*-mers, likely due to homogenization and/or loss of progenitor-specific TEs during long-term diploidization[5,125]. The recently developed Orthology index[46] was employed to identify inter-genomic orthologous relationships (parameter: 'filter -c 0.5'), which showed clearer patterns than *K*s. Two of the three subgenomes of *Du. zibethinus* show 1:1 orthology with the two subgenomes of *F. major* (Fig. 2D, Supplementary Fig. 13D), so the third subgenome (designated as C) of *D. zibethinus* was initially phased out. Then the two sub-genomes (A and B) were differentiated according to their different relationships to the C subgenome. The orthology patterns or *K*s values showed no such differentiation patterns (Supplementary Fig. 13D), so we applied further phylogeny-based evidence (Supplementary Fig. 17). We reconstructed phylogenies for each chromosome of tPMK using WGDI (-at option) and ASTRAL-Pro v1.15.2.3[120], revealing a consistent topology of [A, [B, C]], by defining B as sister to C, and A as sister to B + C clade. In this way, the two subgenomes (A and B) of *F. major* and the three subgenomes (A–C) of *Du. zibethinus* were phased. Then we phased the five subgenomes of *O. pyramidale*. As three of the five subgenomes show a 1:1 orthology with the three subgenomes of *Du. zibethinus* (Fig. 2D, Supplementary Fig. 13G), the three subgenomes were assigned as A, B, and C corresponding to *Du. zibethinus* sub-genomes. These assignments were also validated using the chromo-some phylogenies (Supplementary Fig. 17). The two remaining subgenomes (D and E) were further differentiated based on their phylogenetic relationships to the *R. pubescens* genome. Subgenome D was more closely related to the *R. pubescens* genome based on the chromosome phylogenies. Although subgenome D was not always sister to the *R. pubescens* genome in the consensus trees, we separated subgenomes D and E based on the closer relationship with the *R. pubescens* genome in the gene trees. Then, the A and B subgenomes of *D. trichospermus*, *P. kingtungense* and *C. yunnanensis* were assigned by mapping to the *F. major* genome based on the orthologous relation-ships using WGDI (`-km` option). Both A and B subgenomes of *C. yunnanensis* had two sister copies due to additional paleotetraploidy, and thus were arbitrarily assigned as A1 and A2 and B1 and B2 (relative phasing), respectively. Similarly, subgenomes A–E of *B. ceiba* and *G. raimondii* were assigned by mapping to the *O. pyramidale* genome (WGDI -km). The two subgenomes of *R. pubescens* were also arbitrarily assigned and referred to as D1 and D2 (relative phasing), as further solid evidence for their different genetic origins was lacking, and relative phasing was not expected to affect their sister relationships. Finally, chromosome phylogenies containing all subgenomes were reconstructed for confirmation. The phylogenetic positioning of the A–E subgenome lineages was mostly consistent (i.e. [A, [B, [D, [C, E]]]]), though with some conflicts due to high levels of gene tree discordance (Supplementary Fig. 17). Although the large gene tree discordance can potentially lead to phasing errors between the A–E subgenomes, the discordance may in turn counteract most of the incorrect assignments, as shown by previous benchmarking results[5].

## Reconstruction and timing of subgenome-based phylogeny

Based on the codon alignments from WGDI '-at' option, gene trees were reconstructed using IQ-TREE v2.2.0.3[121] with 1000 bootstraps. A total of 5,194 gene trees, with up to 40% of taxa missing, were then entered into ASTRAL-Hybrid v1.15.2.3[126] to infer the subgenome/spe-cies tree. In addition to the summary method, we concatenated the codon alignments and trimmed the alignment using trimAl (para-meter: -gt 0.8; yielding 1,073,289 sites)[123]. This trimmed alignment was analyzed with IQ-TREE v2.2.0.3[121] to reconstruct the subgenome/spe-cies tree. Furthermore, site-based analyses were conducted using CASTER v1.15.0.0[127] on codon alignments trimmed with trimAl (-gt 0.5; yielding 10,146,564 sites). In addition, we created the split networks using SplitsTree[128] v6.1.16. *V. vinifera* (Vitales), *Ca. papaya* (Brassicales), *A. sinensis* (Malvales; Thymelaeaceae), and *Di. gracilis* (Malvales;

Dipterocarpaceae) were used as outgroups. Based on the results of these different methods, we manually reconciled the phylogenetic topologies of the A and B lineages in the subgenome tree.

We estimated the divergence times between the subgenomes/species based on molecular clocks and fossil calibrations. Three well-documented fossils with confident subfamily affinity were used: (1) The oldest fossil of *Craigia*, a living fossil plant species common during the Tertiary in the Northern Hemisphere, from the Paleocene (66–56 Mya)[129]; (2) the oldest fossil of *Reevesia*, a living fossil plant in the Northern Hemisphere, from the Paleocene (66–56 Mya)[130]; (3) a fossil from a tropical rainforest from the middle to late Paleocene (60–56 Mya), which was assigned to the subfamily Malvoideae based on overall leaf morphology and the identification of shared and derived characters between taxa within this group[131]. The maximum age of the root (the split of grape and cacao) was constrained to 125 Mya according to the TimeTree database[132]. Dating was conducted using two relaxed clock methods, the penalized likelihood method in r8s[133] v1.81 and the Bayesian method in MCMCtree of the PAML package v4.9e[134]. For analysis of r8s, branch lengths (in units of substitutions per site) of the subgenome tree were estimated using IQ-TREE2. For MCMCtree, a concatenated codon alignment using trimAl (-gt 0.9; 79,032 sites) was trimmed to allow at most 10% gaps. The alignment was divided into 3 partitions according to codon positions. The GTR model was used. MCMC chains were sampled 100,000 times every 100 chains, with the initial 10% discarded as burn-in. To confirm the esti-mated times, we also estimated the clade times empirically using CladeDate[135], based exclusively on 141, 129 and 27 fossil records of Malvaceae[136], *Craigia*[129] and *Reevesia*[130]. Finally, the r8s estimates were favored due to their convergence across subgenomes (e.g., consistent time estimates for the divergence of Bombacoideae and Malvoideae subgenomes) and agreement with CladeDate empirical estimates.

## Detection of introgression

Initially, we used PhyTop[137] to visualize the quartet scores (q1, q2, and q3) derived from the ASTRAL analyses for subgenomes (Supplementary Fig. 21). This allowed the calculation of incomplete lineage sorting (ILS) and introgression/hybridization (IH) indices to identify potential intro-gression signals between species/subgenomes where a significant imbalance between q2 and q3 was observed. Conversely, internal nodes that had low values for both indices (roughly ILS-i < 0.3 and IH-i < 0.05) were considered well resolved and interpreted with high confidence. We then used phylogenetic network estimation methods to test the ancient introgression events based on 2020–5120 syntenic gene loci. Given the substantial computational requirements of the methods, we applied a divide-and-conquer strategy by dividing the species/subgenome tree with the well-resolved internal nodes (Supplementary Fig. 21), and we used only one subgenome to represent a well-resolved clade; for example, we used the D1 subgenome of *R. pubescens*, which retained the most syntenic genes, to represent the entire Malvadendrina clade when we tested the gene flow between *M. paniculata*, *T. cacao* and Mal-vadendrina species. *V. vinifera* and *Di. gracilis*[138] served as the outgroups. We inferred the unrooted networks using the fast pseudolikelihood method SNaQ[139] implemented in PhyloNetworks v0.16.3[140]. The number of hybridizations allowed (*h*) was set from 0 to 6, and the best *h* was determined through slope heuristics[139]. Candidate networks that could not be rooted with known outgroups were classified as spurious and therefore discarded. The fit of these candidate networks to the multi-locus data was assessed using a goodness-of-fit test[141]. It is recom-mended to adopt full-likelihood methods to further confirm the nature of the introgression event[142]. Therefore, we used the full-likelihood implementation of the multispecies coalescent with introgression model in BPP v4.7.0[143] to compare the three scenarios (i.e., ghost introgression, inflow and outflow) for species/subgenome trios (Sup-plementary Figs. 24 and 25). The comparison was conducted by calcu-lating the marginal likelihoods for each scenario, with 16 quadrature

points[144]. For each of the 16 MCMC runs, we used 50,000 MCMC iterations as burn-in and then sampled $10^6$ times every 2 iterations. Each run lasted 3–27 days using a single thread. We used a relaxed clock (clock = 2) with the GTR model[145] to address the relatively distant relationships between the studied species. We obtained 4554 and 2291 non-missing genes for the trio [*M. paniculata*, *T. cacao* and *R. pubescens* D1] and the trio [*O. pyramidale* E, *B. ceiba* E and *G. raimondii* E], respectively, and randomly sampled 1000 loci for the BPP runs to reduce computational costs. The first trio yielded marginal likelihoods of −2,990,064, −2,990,057, and −2,989,935 (the highest), while the second trio resulted in −3,068,469 (the highest), −3,068,486, and −3,068,498 for the three introgression scenarios, respectively. The results suggested that the best model for the two trios is Outflow and Ghost introgression, which is consistent with the pseudolikelihood and goodness-of-fit scores (Supplementary Figs. 24 and 25).

### Biased expression analysis
High-throughput sequencing of mRNA (RNA-seq) was performed for leaf tissue (three replicates) of the paleopolyploid species on a DNBSEQ-T7 platform (BGI Inc., Shenzhen, China) in paired-end 150 mode. Reads were cleaned using fastp v0.19.3[146] and then mapped to the reference genome of each species using HISAT2 v2.1.0[147]. Gene expression levels were quantified using transcripts per million reads (TPM) implemented in StringTie v2.1.5[148]. To determine the subgenome dominance for each genome, TPM values were extracted for each set of homoeologs and the relative expression of each homoeolog was calculated by dividing its TPM by the sum of TPMs for the homoeologs. Homoeologs with summed TPM < 1 were discarded.

### Statistical analysis
Gene retention of each subgenome was calculated using the '-r' option of WGDI, with the tPMK as the reference. GO and KEGG function enrichments for retained genes were performed using a hypergeometric method implemented in GOseq package[149].

To test the correlation between the extent of PPD and evolutionary success, we proposed two indices to quantify the extent of PPD in each paleopolyploid genome. The first index, termed chromosome dysploidy rate ($d_C$), quantifies PPD at the chromosome level:

$$d_C = 1 - \frac{x}{N \times p} \tag{1}$$

where $x$ is the extant monoploid chromosome number (e.g. $x = 13$ in the cotton genome), $N$ is the chromosome number of an inferred haploid/monoploid ancestral karyotype of the MRCA of species under study (here we used the constant $N = 11$ of the PMK for all Malvaceae species) and $p$ is the relative ploidy of the monoploid assembly relative to the ancestral karyotype (e.g. $p = 5$ for the cotton genome relative to PMK). The common ancestral chromosome number ($N$) has to be used to compare dysploidy rates across multiple subfamilies.

The second index, termed gene loss rate ($d_G$), quantifies PPD at the gene level:

$$d_G = 1 - \frac{g}{G \times p} \tag{2}$$

where $g$ is the current number of retained syntenic genes (e.g., $g = 25,550$ in the cotton genome), $G$ is the number of ancestral genes before polyploidization (here we used the constant $G = 19,320$ from the outgroup cacao genome for all the species), and $p$ is the relative ploidy (e.g., $p = 5$ for the cotton genome). A higher level of PPD is indicated by a higher $d_C$ or $d_G$.

We used the number of species and genera within each subfamily to quantify evolutionary success. Data were obtained from the World Flora Online (WFO) database (accessed October 2023)[150].

Nucleotide substitution rates (substitutions per site per million years) of each species were estimated by:

$$\mu = \frac{L}{T} \tag{3}$$

where $L$ is the branch length (substitutions per site) from the tip to the crown node of Malvaceae in the ML tree (Fig. 2A) and $T$ is the estimated time of the Malvaceae crown group (i.e., 70 Mya).

Maximum plant height data were compiled from our field surveys, floras (https://www.iplant.cn/), the Plant Trait database[151], and previously published studies[152,153].

The difference between the two samples was assessed using the two-tailed, unpaired Wilcoxon rank-sum test. Correlation analyses comparing two variables were performed under a phylogenetic generalized least squares framework, using the R packages 'ape' 5.0[154] and 'caper' v1.0.3 (https://CRAN.R-project.org/package=caper). For each analysis, the nuclear genomic phylogeny presented in Fig. 2A was used, and Pagel's $\lambda$ is presented to estimate the strength of the phylogenetic signal in the correlational structure of the dataset.

### Reporting summary
Further information on research design is available in the Nature Portfolio Reporting Summary linked to this article.

## Data availability
All the raw genomic sequencing reads were deposited in the Sequence Read Archive (SRA) under Bioproject PRJNA953923 and in the Genome Sequence Archive (GSA) under Bioproject PRJCA019343. The primary monoploid nuclear genome assemblies and annotations were deposited in Genbank under accession GCA_051295405.1 (*D. trichospermus*), GCA_051307295.1 (*P. kingtungense*), GCA_051046955.1 (*C. yunnanensis*), and GCA_050990945.1 (*R. pubescens*). The haplotype-resolved genome assemblies were deposited in Genome Warehouse (GWH) under accession GWHESEP00000000 (*D. trichospermus*), GWHESEQ00000000 (*P. kingtungense*), GWHESER00000000 (*C. yunnanensis*), and GWHESES00000000 (*R. pubescens*). The plastid and mitochondrion genome assemblies and annotations were deposited in Genbank under accession OR832745–OR832754 and OR839124–OR839137. The files of the predicted ancestral karyotypes were deposited in Figshare [https://doi.org/10.6084/m9.figshare.29043869]. Source data are provided with this paper.

## Code availability
The codes supporting our methodology can be found at GitHub [https://github.com/zhangrengang/MAL].

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

## Acknowledgements

We thank Dr. Jing Yang, Prof. Wei-Bang Sun, Prof. Bin Tian, and Dr. Guo-Fang Yuan for sharing their unpublished or updated genomic data. We thank Dr. Jie Cai, Heng Shu, and Hong-Hao Jiang for their help in sampling. We thank Drs. Wei Zhao, Li-Dan Tao, Yan Liu, Profs. SiuYi Li, Qin Qiao and Ti-Cao Zhang for helpful discussions. We thank Prof. Wen-Bin Yu for taxonomic suggestions. We also thank Drs. Peng-Chuan Sun and Chao Zhang for assistance using WGDI and ASTRAL, respectively. The work was equally supported by the National Natural Science Foundation of China (32471734) and the National Key Research and Development Program (2024YFF1307400), funded in part by the "Light of West China" Program and the project TowArdsNext GENeration Crops (CZ.02.01.01/00/22_008/0004581) of the ERDF Programme Johannes Amos Comenius.

## Author contributions

Y.P.M., D.W.L., X.Y.G., M.A.L., J.F.W., and R.G.Z. conceived and designed the study; D.T.L., S.C.S., X.F.L., H.Z., and Y.P.M. sampled the plants; Y.P.M., H.Z., M.M.L., C.Y.J., Y.H.L., and X.Y.S. performed experiments; R.G.Z., H.Y.S., M.J.Z., H.Z., and J.L.C. analyzed the data; R.G.Z., H.Z., H.Y.S., and J.L.C. prepared figures; R.G.Z., H.Z., and D.W.L. wrote the paper; Y.P.M., M.A.L., J.L.C., J.F.W., D.W.L., and K.H.J. revised the paper; all authors approved the final paper.

## Competing interests

The authors declare no competing interests.

## Additional information

¹Yunnan Key Laboratory for Integrative Conservation of Plant Species with Extremely Small Populations/State Key Laboratory of Plant Diversity and Specialty Crops, Kunming Institute of Botany, Chinese Academy of Sciences, Kunming 650201, China. ²University of Chinese Academy of Sciences, Beijing 101408, China. ³State Key Laboratory of Cotton Biology, Institute of Cotton Research, Chinese Academy of Agricultural Sciences, Anyang 455000, China. ⁴College of Life Sciences, Qufu Normal University, Qufu 273165, China. ⁵Ecology and Evolutionary Biology Department/Molecular and Cellular Biology Department, University of Arizona, Tucson, AZ 85718, USA. ⁶Donald Danforth Plant Science Center, St. Louis, MO 63132, USA. ⁷Institute of Crop Germplasm Resources, Shandong Academy of Agricultural Sciences, Jinan 250100, China. ⁸CAS Key Laboratory of Tropical Forest Ecology, Xishuangbanna Tropical Botanical Garden, Chinese Academy of Sciences, Mengla 666303, China. ⁹Wuhan Botanical Garden, Chinese Academy of Sciences, Wuhan 430074, China. ¹⁰CEITEC—Central European Institute of Technology and Department of Experimental Botany, Faculty of Science, Masaryk University, Brno CZ-625 00, Czech Republic. ¹¹Department of Ecology, Evolution, and Organismal Biology, Iowa State University, Amees, IA 50011, USA. ¹²These authors contributed equally: Ren-Gang Zhang, Hang Zhao, Justin L. Conover, Hong-Yun Shang ✉e-mail: lidawei@wbgcas.cn; martin.lysak@ceitec.muni.cz; jfw@iastate.edu; gexiaoyang@caas.cn; mayongpeng@mail.kib.ac.cn

