## [Peer Review file · Nature Communications]

Reticulate allopolyploidy and subsequent dysploidy drive evolution and diversification in the cotton family

Corresponding Author: Professor Yongpeng Ma

Version 0:

Reviewer comments:

Reviewer #1

(Remarks to the Author)

This ms presents a comprehensive genomic investigation into the evolutionary history of the Malvaceae family, leveraging high-quality chromosome-scale genomes to elucidate the roles of polyploidy and post-polyploid diploidization (PPD) in driving diversification. The authors provide compelling evidence for the "polyploidy for survival, PPD for success" hypothesis, supported by robust correlations between dysploidy rates and taxonomic richness. The identification of the completely retained COL9 gene family as a regulator of flowering time adds functional relevance to the genomic findings. The work is methodologically rigorous, integrating cutting-edge sequencing, subgenome analysis, and phylogenetic network approaches. This study significantly advances our understanding of plant genome evolution and offers valuable insights into the genomic basis of biodiversity in angiosperms. However, some concerns should be addressed as below.

Major comments

While the study includes genomes from all nine Malvaceae subfamilies, several subfamilies (e.g., Brownlowioideae, Dombeyoideae) are represented by only one species. This raises concerns about whether the observed patterns generalize across each subfamily's diversity. For example, the Bombacoideae and Malvoideae comparisons rely on two and one species, respectively, which may not fully capture intra-subfamilial variation. The authors should discuss how limited sampling might affect their conclusions, particularly regarding the correlation between dysploidy rates and diversification.

The study identifies "anomaly zones" with high incomplete lineage sorting (ILS) and introgression signals, leading to unresolved relationships in subgenome phylogenies. While the authors acknowledge this, the implications of these uncertainties for reconstructing reticulate polyploidization events (e.g., Mal- α and Mal- β) remain unclear. Additional analyses, such as alternative network methods (e.g., PhyloNet), may strengthen confidence in the proposed evolutionary trajectories.

The role of COL9 in delaying flowering is demonstrated in *Gossypium hirsutum*, but its conservation and function across Malvaceae are not explored. Given the family-wide retention of COL9 homologs, functional assays in other subfamilies would clarify whether this gene family universally influences photoperiodic adaptation. At least, the authors should discuss the limitations of extrapolating COL9's role from cotton to the entire family. Moreover, the study emphasizes retained COL9 homologs but does not address whether retention reflects selection or neutral processes. A dN/dS analysis across Malvaceae lineages could test for signatures of purifying selection, supporting the hypothesis that COL9 retention is adaptive.

Minor comments

- Figure 1A: The assembly metrics (e.g., N50, BUSCO) are critical but presented in a dense table. A visual summary (e.g., bar plots) would improve readability.

- The term "reticulate polyploidization" (line 241) is used broadly. Distinguishing between autopolyploidy and allopolyploidy in different events would refine the narrative.

- The Wilcoxon tests for gene retention (lines 373-375) should specify whether paired or unpaired tests were used, as this affects interpretation.

- The proposed "four-step" decaploidization model (lines 378-383; Fig. S43) is speculative without intermediate (hexaploid/octoploid) genomes. The authors should acknowledge that this as a hypothesis.
- The k-mer analysis (Suppl. Methods, line 134) mentions "KMC v3.1.1" but omits parameters (e.g., k-mer size). Specify settings (e.g., "k=21") to enable replication.
- Subgenome phasing is central to the analysis but described briefly. Expanding this section (Suppl. Methods), including criteria for subgenome assignments (e.g., Orthology Index thresholds), would enhance reproducibility.
- The dysploidy rate formula assumes a constant ancestral chromosome number ($N = 11$). Justification for this assumption (e.g., references to prior studies) should be provided.
- Supplementary Note 4: "MCRA" is undefined; spell out.
- Figures S11–S14 all describe synteny dot plots but lack distinct captions explaining their unique contributions. Differentiate each figure's purpose (e.g., S11: inter-subfamily synteny; S12: intra-tetraploid comparisons).
- Fig. S42A indicates that there is also biased fractionation between the subgenomes generated by $Til-\alpha$ and $Hel-\beta$, implying that these WGD events may also be allopolyploidy. The authors should consider this possibility and discuss whether these WGD events involved hybridization between different genomes.
- Figure S44A highlights disrupted expression trends in cotton subgenomes. This anomaly warrants discussion—e.g., whether domestication or recent selection in cotton altered ancestral expression patterns.
- Data availability: The reconstructed ancestral karyotypes are a core result of the study and should be made publicly available. Consider depositing these data on repositories such as Figshare to enhance transparency and allow other researchers to verify and build upon the findings.

Overall, this work represents a landmark contribution to plant genomics, offering novel insights into the interplay of polyploidy, diploidization, and diversification. Addressing the major concerns above would solidify the study's conclusions and broaden its impact.

Reviewer #2

(Remarks to the Author)

I really enjoyed this paper, it is extremely interesting and sheds much light on the patterns of genome and chromosome divergence in Malvaceae. In addition, and with great novelty, it reveals the importance of polyploidy and critically, post polyploid diploidisation (PPD) in driving the group's diversity. The work's insights into PPD adds significant value to the paper's importance.

This paper is a very carefully constructed, complex manuscript, that is beautifully presented. Figure 3 is a super summary of events.

My only suggestion, and it is not essential that the authors act upon this suggestion, is that the flowering gene analysis is published separately. The central narrative and novelty of the paper is the genome level analysis and interpretation of polyploidy and PPD. This alone has five figures and 46 supplementary figures and it could stand alone. The flowering gene narrative, with its two main figures and 6 supplementary figures is not a perfect fit, and is sufficient to be published in its own right.

Minor suggestions

- Paragraph starting line 239. If the authors think it is appropriate, it might be worth mentioning the work of Weiss Schneeweiss, Parker and colleagues (doi: 10.3389/fpls.2018.00433 or papers over two preceding decades) who use cytogenetics to show much reticulate evolution in polyploids of *Prospero autumnale* complex.
- The Y axis colour distinctions of chromosomes depicted on Fig. S38A is poor (although resolvable) and could be improved.
- Line 401. Building on the sentence 'The formation and diversification of these paleopolyploid genomes around the K–Pg boundary and predating the PETM (Fig. 4C) support the link between WGDs and survival of flowering plants during the environmental catastrophes', the authors could potentially speculate that the establishment of the polyploid ancestors may have been facilitated by the vacant niches opened up by the K-Pg extinctions.
- Fig. 1 legend the authors write (A) 'The confidently well-resolved nodes are indicated by pie charts with the frequencies of three gene tree topologies (q_1 , q_2 and q_3 ; $q_1 \gg q_2/q_3$) calculated in ASTRAL'. However, I don't see any resolution inside the pie charts, and certainly nothing like that shown in Fig S17. (D), the phrase is repeated 'are colored according to the Orthology Index'.
- Figure 3, Despite the figure being an excellent summary of their data, I was initially delayed in my understanding of its message by two similar boxes, one for 'ploidy level' and one for 'Dysploidy /CR'. Can these boxes be better differentiated

(perhaps by colour)? In comparing the text narrative with the figure, we learn that 2x in a square box means paleotetraploid, 3x paleohexaploid etc. Perhaps the legend could state that definition explicitly. The pale green for 'Paleopolyploidy events' is very indistinct (at least to me).

Version 1:

Reviewer comments:

Reviewer #1

(Remarks to the Author)

Excluding the following issues, the authors have adequately addressed the previous review comments. The authors cite a literature reference indicating that a single species can represent the mean chromosome number of a subfamily (Lines 505-513). It is suggested that the authors provide a specific statistical table of chromosome counts to demonstrate that the selected species' chromosome number is indeed not significantly different from the overall subfamily.

Reviewer #2

(Remarks to the Author)

This manuscript is improved from an already good starting point following referee comments. I am pleased they chose to remove the flowering gene analysis, keeping the manuscript focussed on the novelty of their insights and findings related to post polyploidy diploidisation, which is interesting important and novel, adding another dimension to their through and interesting analysis of genome divergence in the Malvaceae.

I have one very minor query in the legend to figure 3. line 784. Can the authors confirm line 784 'The circled numbers show the haploid and monoploid chromosome numbers of the ancestral...' They present here one number for two entities, I think they mean only haploid?

Version 2:

Reviewer comments:

Reviewer #1

(Remarks to the Author)

The author has diligently addressed the raised questions, making thorough revisions and providing detailed responses. Currently, this version does not require any additional modifications.

Reviewer #2

(Remarks to the Author)

The authors have addressed my minor comment. The work is interesting and important.

Response to REVIEWER COMMENTS

Reviewer #1 (Remarks to the Author):

This ms presents a comprehensive genomic investigation into the evolutionary history of the Malvaceae family, leveraging high-quality chromosome-scale genomes to elucidate the roles of polyploidy and post-polyploid diploidization (PPD) in driving diversification. The authors provide compelling evidence for the "polyploidy for survival, PPD for success" hypothesis, supported by robust correlations between dysploidy rates and taxonomic richness. The identification of the completely retained COL9 gene family as a regulator of flowering time adds functional relevance to the genomic findings. The work is methodologically rigorous, integrating cutting-edge sequencing, subgenome analysis, and phylogenetic network approaches. This study significantly advances our understanding of plant genome evolution and offers valuable insights into the genomic basis of biodiversity in angiosperms. However, some concerns should be addressed as below.

Response: We thank to the Reviewer for positive and constructive comments. We have carefully considered all suggestions in order to improve the manuscript.

Major comments

While the study includes genomes from all nine Malvaceae subfamilies, several subfamilies (e.g., Brownlowioideae, Dombeyoideae) are represented by only one species. This raises concerns about whether the observed patterns generalize across each subfamily's diversity. For example, the Bombacoideae and Malvoideae comparisons rely on two and one species, respectively, which may not fully capture intra-subfamilial variation. The authors should discuss how limited sampling might affect their conclusions, particularly regarding the correlation between dysploidy rates and diversification.

Response: To address this concern, we have added a detailed discussion in the revised manuscript (see Lines 505-513) regarding the impact of limited species representation on our conclusions. While limited sampling may not fully capture intra-subfamilial variation, we believe its effect on our conclusions about the correlation between dysploidy rates and diversification is minimal, as the representative species have chromosome numbers close to the mean or modal chromosome number values of their respective subfamilies. Nonetheless, we acknowledge that limited sampling may obscure finer-scale variation in dysploidy rates within subfamilies with a high species diversity. We also acknowledge that future studies with expanded taxon sampling will further validate the pattern observed in our study.

The study identifies "anomaly zones" with high incomplete lineage sorting (ILS) and introgression signals, leading to unresolved relationships in subgenome phylogenies. While the authors acknowledge this, the implications of these uncertainties for reconstructing reticulate polyploidization events (e.g., Mal- α and Mal- β) remain unclear. Additional analyses, such as alternative network methods (e.g., PhyloNet), may strengthen confidence in the proposed evolutionary trajectories.

Response: To address the concern on "anomaly zones", we have expanded our discussion in the revised manuscript (see discussion in **Supplementary Note 5**) to clarify the impact of these

uncertainties on reconstructing reticulate polyploidization events (e.g., Mal- α and Mal- β). Below, we outline why these uncertainties have limited impact on most of our conclusions and where they may influence specific inferences.

The following conclusions regarding reticulate polyploidization events are robust and not affected by “anomaly zones”:

- 1) Presence and resulting ploidy of polyploidization events: These are supported by clear synteny patterns with outgroups and distinct Ks peaks, independent of gene tree conflicts (**Figs. 2B–C, S11**).
- 2) Shared vs. independent polyploidization events across subfamilies: Consistent evidence from orthologous synteny and phylogenetic analyses supports these patterns, not affected by anomaly zones (**Figs. 2D, S12–S14, S17–S19**).

“Anomaly zones” may affect inferences for the following questions, but the inferences are supported by consistent evidence from multiple methods:

- 1) Progenitor of the hexaploid *Du. zibethinus*: The A/B subgenomes of the hexaploid are consistently sister to those of *F. major* across all phylogenetic trees (**Fig. S17–S19**), robustly identifying *F. major* (or its close relatives) as the tetraploid progenitor despite the gene tree discordance.
- 2) Progenitor relationship between the hexaploid and the decaploid: The A/B subgenomes of the hexaploid *D. zibethinus* are not sister to those of the decaploid in any phylogenetic trees (**Fig. S17–S19**), confirming that the hexaploid (or its sisters) is not the decaploid’s direct progenitor. These conclusions hold across chromosomes and methods, mitigating the impact of the gene tree discordance.

To address the reviewer’s suggestion, we tested PhyloNet’s InferNetwork_MPL command to resolve relationships within the anomaly zones. However, the resulting networks also did not pass the goodness-of-fit assessments, likely due to the extreme biological complexity in these zones, including short internal branch lengths, high ILS, rampant introgression, and long-term gene fractionation over ~60 million years. These factors suggest rapid species radiation coupled with extensive gene flow, making precise resolution challenging. While split networks have provided valuable insights into these complex relationships, we acknowledge the limitations of current methods to fully resolve anomaly zones and we discuss the issue in the revised manuscript version (**Supplementary Note 5**). Future advances in network inference methods may offer opportunities to further clarify these relationships.

The role of COL9 in delaying flowering is demonstrated in *Gossypium hirsutum*, but its conservation and function across Malvaceae are not explored. Given the family-wide retention of COL9 homologs, functional assays in other subfamilies would clarify whether this gene family universally influences photoperiodic adaptation. At least, the authors should discuss the limitations of extrapolating COL9’s role from cotton to the entire family. Moreover, the study emphasizes retained COL9 homologs but does not address whether retention reflects selection or neutral

processes. A dN/dS analysis across Malvaceae lineages could test for signatures of purifying selection, supporting the hypothesis that COL9 retention is adaptive.

Response: Following Reviewer #2's suggestion, we have removed the section on COL9 genes to streamline the manuscript, and focused more on the genome evolution story. We appreciate your insights and plan to explore COL9's conservation and function more deeply in future work.

Minor comments

- Figure 1A: The assembly metrics (e.g., N50, BUSCO) are critical but presented in a dense table. A visual summary (e.g., bar plots) would improve readability.

Response: We have visualized the assembly metrics in **Fig. 1A**.

- The term "reticulate polyploidization" (line 241) is used broadly. Distinguishing between autopolyploidy and allopolyploidy in different events would refine the narrative.

Response: We have specified the term as "reticulate allopolyploidization" throughout the revision (e.g., line 243).

- The Wilcoxon tests for gene retention (lines 373-375) should specify whether paired or unpaired tests were used, as this affects interpretation.

Response: We have specified them as unpaired tests in the Method section and figure legend.

- The proposed "four-step" decaploidization model (lines 378-383; Fig. S43) is speculative without intermediate (hexaploid/octoploid) genomes. The authors should acknowledge that this as a hypothesis.

Response: We have explicitly acknowledged it as a hypothesis (line 403 and **Fig. S43**).

- The k-mer analysis (Suppl. Methods, line 134) mentions "KMC v3.1.1" but omits parameters (e.g., k-mer size). Specify settings (e.g., "k=21") to enable replication.

Response: We have provided the parameters in the revision (Suppl. Methods, line 149).

- Subgenome phasing is central to the analysis but described briefly. Expanding this section (Suppl. Methods), including criteria for subgenome assignments (e.g., Orthology Index thresholds), would enhance reproducibility.

Response: We have added both the criteria and Code availability to make the methodology more clear (lines 534-536).

- The dysploidy rate formula assumes a constant ancestral chromosome number ($N = 11$). Justification for this assumption (e.g., references to prior studies) should be provided.

Response: As this dysploidy rate formula is newly proposed in our study, no prior studies directly support this specific approach. The assumption of a constant ancestral chromosome number ($N = 11$) is based on the need to establish a common baseline for comparing dysploidy rates across multiple subfamilies, using the inferred chromosome number of their most recent common ancestor (MRCA). To clarify this rationale, we have added a brief explanation in the revised manuscript (see Suppl. Methods, lines 412-413).

- Supplementary Note 4: "MCRA" is undefined; spell out.

Response: Done.

- Figures S11–S14 all describe synteny dot plots but lack distinct captions explaining their unique contributions. Differentiate each figure's purpose (e.g., S11: inter-subfamily synteny; S12: intra-tetraploid comparisons).

Response: Done.

- Fig. S42A indicates that there is also biased fractionation between the subgenomes generated by *Til- α* and *Hel- β* , implying that these WGD events may also be allopolyploidy. The authors should consider this possibility and discuss whether these WGD events involved hybridization between different genomes.

Response: We have added this possibility (lines 409-411).

- Figure S44A highlights disrupted expression trends in cotton subgenomes. This anomaly warrants discussion—e.g., whether domestication or recent selection in cotton altered ancestral expression patterns.

Response: We have added discussion for this anomaly (lines 408-409).

- Data availability: The reconstructed ancestral karyotypes are a core result of the study and should be made publicly available. Consider depositing these data on repositories such as Figshare to enhance transparency and allow other researchers to verify and build upon the findings.

Response: We have deposited these data on Figshare (DOI: 10.6084/10.6084/m9.figshare.29043869 ; reviewer link: <https://figshare.com/s/05bb64644235f64784c0>) (lines 530-532).

Overall, this work represents a landmark contribution to plant genomics, offering novel insights into the interplay of polyploidy, diploidization, and diversification. Addressing the major concerns above would solidify the study's conclusions and broaden its impact.

Response: Thanks again.

Reviewer #2 (Remarks to the Author):

I really enjoyed this paper, it is extremely interesting and sheds much light on the patterns of genome and chromosome divergence in Malvaceae. In addition, and with great novelty, it reveals the importance of polyploidy and critically, post polyploid diploidisation (PPD) in driving the group's diversity. The work's insights into PPD adds significant value to the paper's importance.

This paper is a very carefully constructed, complex manuscript, that is beautifully presented. Figure 3 is a super summary of events.

Response: Thanks for your positive and constructive comments. We have followed your suggestions to improve the manuscript.

My only suggestion, and it is not essential that the authors act upon this suggestion, is that the flowering gene analysis is published separately. The central narrative and novelty of the paper is the genome level analysis and interpretation of polyploidy and PPD. This alone has five figures and 46 supplementary figures and it could stand alone. The flowering gene narrative, with its two main figures and 6 supplementary figures is not a perfect fit, and is sufficient to be published in its own right.

Response: We thank the reviewer for the suggestion to streamline the manuscript's focus. To align with the core narrative of genome-level polyploidy and PPD analysis, we have removed the flowering gene analysis, including its two main figures and six supplementary figures, from the revised manuscript (see revised Results and Supplementary Materials). The current results of COL9 plus future gene validation work will be separately published.

Minor suggestions

- Paragraph starting line 239. If the authors think it is appropriate, it might be worth mentioning the work of Weiss Schneeweiss, Parker and colleagues (doi: 10.3389/fpls.2018.00433 or papers over two preceding decades) who use cytogenetics to show much reticulate evolution in polyploids of *Prospero autumnale* complex.

Response: We have cited the work (10.3389/fpls.2018.00433) in the revision (line 244).

- The Y axis colour distinctions of chromosomes depicted on Fig. S38A is poor (although resolvable) and could be improved.

Response: We have changed the color on Fig. S38A to improve the distinctions.

- Line 401. Building on the sentence 'The formation and diversification of these paleopolyploid genomes around the K–Pg boundary and predating the PETM (Fig. 4C) support the link between WGDs and survival of flowering plants during the environmental catastrophes', the authors could potentially speculate that the establishment of the polyploid ancestors may have been facilitated by the vacant niches opened up by the K-Pg extinctions.

Response: We have added the speculation (lines 427-428).

- Fig. 1 legend the authors write (A) 'The confidently well-resolved nodes are indicated by pie charts with the frequencies of three gene tree topologies (q_1 , q_2 and q_3 ; $q_1 \gg q_2/q_3$) calculated in ASTRAL'. However, I don't see any resolution inside the pie charts, and certainly nothing like that shown in Fig S17. (D), the phrase is repeated 'are colored according to the Orthology Index'.

Response: We have enlarged the pie charts to make them clearer. However, the primary reason for the comment "don't see any resolution inside the pie charts, and certainly nothing like that shown in Fig. S17" is that the q_1 values for these well-resolved nodes are much greater than q_2/q_3 (indicating minimal gene tree conflicts), resulting in the q_2/q_3 proportions being too small to be well visible. Our aim here is to demonstrate that these three nodes (i.e. the crown nodes of Malvaceae, Malvadendrina and Malvatheca) are highly robust, requiring minimal further attention, while the remaining nodes warrant additional analysis. Fig. S10 presents the pie charts for the remaining

nodes, which were not shown in Fig. 2A due to their large number. We have referred to Fig. S10 in the caption for better visualization.

The repeated phrase has been deleted.

- Figure 3, Despite the figure being an excellent summary of their data, I was initially delayed in my understanding of its message by two similar boxes, one for 'ploidy level' and one for 'Dysploidy /CR'. Can these boxes be better differentiated (perhaps by colour)? In comparing the text narrative with the figure, we learn that 2x in a square box means paleotetraploid, 3x paleohexaploid etc. Perhaps the legend could state that definition explicitly. The pale green for 'Paleopolyploidy events' is very indistinct (at least to me).

Response: We have differentiated these boxes by color.

We have added annotation in the legend ('relative ploidy') and caption ('e.g., 2×, indicating paleotetraploidy, and 5×, paleodecaploidy').

We have replaced the color for 'Paleopolyploidy events', ensuring better visibility and contrast.

Please check Fig. 3 again.

Response to REVIEWER COMMENTS

Reviewer #1 (Remarks to the Author):

Excluding the following issues, the authors have adequately addressed the previous review comments

The authors cite a literature reference indicating that a single species can represent the mean chromosome number of a subfamily (Lines 505-513). It is suggested that the authors provide a specific statistical table of chromosome counts to demonstrate that the selected species' chromosome number is indeed not significantly different from the overall subfamily.

Response: Thanks again for your constructive comment. To address your concern, we compared the chromosome numbers of the selected species with those of the overall subfamily (Supplementary Fig. 48). Our results reveal that the chromosome numbers of selected species were not significantly different from the overall subfamily ($P > 0.05$) but were close to the subfamily means or medians, only except for the Helicteroideae, based on data from the literature (Marinho et al., 2014). These results indicate the representativeness of subfamily chromosome numbers using the selected species, which has been presented in Supplementary Fig. 48 (see also below).

Supplementary Fig. 48. Comparison of chromosome counts of the selected species with those of the overall subfamily, based on data from Marinho et al. (2014). *, $P \leq 0.05$; ns, $P > 0.05$; two-tailed Wilcoxon rank-sum test.

Reviewer #2 (Remarks to the Author):

This manuscript is improved from an already good starting point following referee comments. I am pleased they chose to remove the flowering gene analysis, keeping the manuscript focussed on the novelty of their insights and findings related to post polyploidy diploidisation, which is interesting

important and novel, adding another dimension to their thorough and interesting analysis of genome divergence in the Malvaceae.

Response: Thanks again for your positive comments.

I have one very minor query in the legend to figure 3. line 784. Can the authors confirm line 784 'The circled numbers show the haploid and monoploid chromosome numbers of the ancestral...!' They present here one number for two entities, I think they mean only haploid?

Response: We confirm that the circled numbers represent the haploid chromosome number (n) for ancestral genomes and the monoploid chromosome number (x) for modern genomes, respectively. The numbers do not represent one value for two entities. The use of “monoploid” for modern genomes is necessary due to neoautopolyploidy in some species, such as *Craigia yunnanensis* ($2n = 4x = 164$), where the monoploid number (x) reflects the basic chromosome set, unaffected by neoautopolyploidy. To address your concern and enhance clarity, we have revised the sentence to: “The circled numbers indicate either the haploid chromosome number for ancestral genomes or the monoploid chromosome number for modern genomes”.

We hope this clarifies the distinction and addresses your query.